# Defining the interactome of the human mitochondrial ribosome identifies SMIM4 and TMEM223 as respiratory chain assembly factors

Sven Dennerlein[1†], Sabine Poerschke[1†], Silke Oeljeklaus[2,3], Cong Wang[1], Ricarda Richter-Dennerlein[1,4], Johannes Sattmann[1], Diana Bauermeister[1], Elisa Hanitsch[1], Stefan Stoldt[4,5,6], Thomas Langer[7], Stefan Jakobs[4,5,6,8], Bettina Warscheid[2,3], Peter Rehling[1,4,8,9]*

[1]Department of Cellular Biochemistry, University Medical Center Göttingen, Göttingen, Germany; [2]Biochemistry and Functional Proteomics, Institute of Biology II, University of Freiburg, Freiburg, Germany; [3]Signalling Research Centres BIOSS and CIBSS, University of Freiburg, Freiburg, Germany; [4]Cluster of Excellence "Multiscale Bioimaging: from Molecular Machines to Networks of Excitable Cells" (MBExC), University of Göttingen, Göttingen, Germany; [5]Department of NanoBiophotonics, Max Planck Institute for Biophysical Chemistry, Göttingen, Germany; [6]Department of Neurology, University Medical Center Göttingen, Göttingen, Germany; [7]Department of Mitochondrial Proteostasis, Max Planck Institute for Biology of Ageing, Cologne, Germany; [8]Fraunhofer Institute for Translational Medicine and Pharmacology ITMP, Translational Neuroinflammation and Automated Microscopy, Göttingen, Germany; [9]Max Planck Institute for Biophysical Chemistry, Göttingen, Germany

**\*For correspondence:**
peter.rehling@medizin.uni-goettingen.de

†These authors contributed equally to this work

**Competing interest:** The authors declare that no competing interests exist.

**Abstract** Human mitochondria express a genome that encodes thirteen core subunits of the oxidative phosphorylation system (OXPHOS). These proteins insert into the inner membrane co-translationally. Therefore, mitochondrial ribosomes engage with the OXA1L-insertase and membrane-associated proteins, which support membrane insertion of translation products and early assembly steps into OXPHOS complexes. To identify ribosome-associated biogenesis factors for the OXPHOS system, we purified ribosomes and associated proteins from mitochondria. We identified TMEM223 as a ribosome-associated protein involved in complex IV biogenesis. TMEM223 stimulates the translation of COX1 mRNA and is a constituent of early COX1 assembly intermediates. Moreover, we show that SMIM4 together with C12ORF73 interacts with newly synthesized cytochrome *b* to support initial steps of complex III biogenesis in complex with UQCC1 and UQCC2. Our analyses define the interactome of the human mitochondrial ribosome and reveal novel assembly factors for complex III and IV biogenesis that link early assembly stages to the translation machinery.

## Editor's evaluation

In this work, the authors analyze the interactome of the human mitochondrial ribosomes and identify two new mitochondrial inner membrane proteins, TMEM223 and SMIM4, as ribosome-associated proteins involved in the biogenesis of respiratory chain complexes. The study reveals novel assembly factors for complex III and IV biogenesis that link early assembly stages to the mitochondrial translation machinery.

## Introduction

Mitochondria play key roles in a plethora of cellular processes such as signaling processes, metabolism, and energy production (*Pfanner et al., 2019*). Among this multitude of functions, cellular energy conversation by oxidative phosphorylation (OXPHOS) is a hallmark. ATP is produced by the mitochondrial OXPHOS system, which is comprised of the respiratory chain complexes I–IV and the $F_1F_o$-ATP-synthase (complex V). Except for complex II, these multi-subunit complexes in the inner mitochondrial membrane (IMM) are composed of nuclear- and mitochondrial-encoded proteins.

The human mitochondrial genome (mtDNA) encodes 2 rRNAs, 22 tRNAs, and 13 proteins. These proteins are synthesized by membrane-associated mitochondrial ribosomes (mt-ribosome) to insert their translation products into the membrane co-translationally (*Englmeier et al., 2017*; *Pfeffer et al., 2015*). Subsequently, these subunits have to engage with nuclear-encoded, imported subunits to form functional enzyme machineries. This process requires a large number of chaperone-like assembly factors, which promote the maturation of the complexes through a number of assembly intermediates.

Since the accumulation of non-assembled OXPHOS proteins or subcomplexes can lead to the production of cellular damaging radicals (e.g., ROS), their assembly processes are highly regulated in a stoichiometrically and temporally manner, which is facilitated by a plethora of assembly factors. These biogenesis factors act at different maturation stages and stabilize assembly intermediates, insert cofactors or ensure the correct protein membrane integrity.

The assembly pathway of the cytochrome *c* oxidase is one of the best characterized processes (*Dennerlein et al., 2017*; *Timón-Gómez et al., 2018*). The three core proteins COX1, COX2, and COX3 are encoded by the mtDNA. COX1 represents the step-stone of the assembly pathway, while COX2 and COX3 get added in a sequential manner. In the yeast *Saccharomyces cerevisiae* (*S. cerevisiae*), COX1 translation is regulated by translational activators that bind to the 5′ untranslated region (UTR) of COX1 mRNA. However, due to the lack of a significant 5′UTR of human COX1 mRNA, such a mechanism probably does not exist in human mitochondria. It has been recently shown that COX1 translation is regulated by early and late assembly stages. MITRAC (mitochondrial translation regulation assembly intermediate of the cytochrome *c* oxidase) represents the COX1 specific assembly intermediate, which comprises at least two sub-complexes (MITRAC[early] and MITRAC[late]) (*Mick et al., 2012*; *Richter-Dennerlein et al., 2016*). MITRAC[early], which interacts directly with the mt-ribosome during COX1 synthesis, is considered as the COX1 translation regulation complex, containing C12ORF62 (COX14) and MITRAC12 (COA3) (*Richter-Dennerlein et al., 2016*). In MITRAC[late], the first nuclear-encoded subunit, COX4I1, joins the assembly intermediate impairing the accomplishment of COX1 synthesis. Thus, the transition from MITRAC[early] to MITRAC[late] represents a key regulatory step for COX1 synthesis and downstream events during cytochrome *c* oxidase biogenesis. However, how the MITRAC complexes regulate COX1 translation on a molecular level remains unclear.

Other mitochondrial OXPHOS assembly pathways, such as for the cytochrome *c* reductase (complex III), have been predominantly investigated in *S. cerevisiae*. The biogenesis of the cytochrome *c* reductase relies on the expression and coordinated assembly of 10 nDNA-encoded subunits and one mtDNA-encoded subunit, cytochrome *b* (CytB). The assembly process starts with the translation of CytB. During synthesis, two translation factors are bound to the nascent polypeptide emerging at the exit tunnel—Cbp3 (UQCC1) and Cbp6 (UQCC2), forming intermediate 0 (*Fernandez-Vizarra and Zeviani, 2018*; *Ndi et al., 2018*). Both translation factors mediate the insertion of newly synthesized CytB into the IMM and dissociate once synthesis is complete. After incorporation of the first haem-b (bL), a third factor—Cbp4 (UQCC3) joins the pre-complex and the second heam bH gets integrated (intermediate I) (*Fernandez-Vizarra and Zeviani, 2018*; *Ndi et al., 2018*). The release of Cbp3/Cbp6 (UQCC1/UQCC2) from the fully hemylated CytB is triggered by the insertion of the structural subunits Qcr7 (UQCRB) and Qcr8 (UQCRQ). Now dimerization occurs and the Cor1/Cor2 modules are joining (*Stephan and Ott, 2020*). In yeast, the translation activators Cbp3/Cbp6 are now available to initiate a new translation cycle of CytB (intermediate II) (*Fernandez-Vizarra and Zeviani, 2018*; *Ndi et al., 2018*). The addition of Rip1 (UQCRFS1) together with the smallest subunit Qcr10 (UQCR11) to a dimeric subcomplex (pre-cIII2) is a crucial maturation step and ensures its catalytic activity. The composition of the cytochrome c reductase from yeast to human is highly similar, where cytochrome c1 (CYC1), Rip1 (UQCRFS1), and CytB (CYTB) form the core, which is organized in a tightly bound symmetrical dimer.

To define the interplay of translation and assembly of mitochondrial OXPHOS complexes, we defined the interactome of the human mt-ribosome under mild solubilization conditions. Among the identified proteins, we detected the uncharacterized protein TMEM223 and showed that it is involved in cytochrome *c* oxidase assembly. Furthermore, we identified SMIM4, which interacts with the recently described cytochrome *c* reductase assembly factor C12ORF73 (*Zhang et al., 2020*). We demonstrate that both proteins are involved in cytochrome *c* reductase biogenesis and that their interplay links mitochondrial translation to cytochrome *c* reductase assembly.

## Results
### Identification of TMEM223 and SMIM4 as mt-ribosome-associated proteins

At the IMM, the mt-ribosome synthesizes 13 essential OXPHOS subunits in human. During translation, it associates with the OXA1L insertase and early assembly factors of the OXPHOS system (*Itoh et al., 2021*). To identify factors that are associated with the mt-ribosome and thereby contribute to OXPHOS biogenesis, we generated a human HEK293T cell line that enables inducible expression of a FLAG-tagged version of the ribosomal subunit mL62$^{FLAG}$ (*Richter-Dennerlein et al., 2016*). mL62 is a component of the 39S large ribosomal subunit (39S mtLSU) (*Richter et al., 2010*; *Brown et al., 2014*; *Greber et al., 2014*; *Busch et al., 2019*). To capture interactions of proteins with the mitochondrial translation machinery, we isolated mitochondria and performed co-immunoisolation experiments under mild solubilization conditions. Proteins that co-purified with mL62$^{FLAG}$ were subjected to sucrose gradient centrifugation and gradient fractions analyzed by western blotting. As expected, the gradient distribution of mL62$^{FLAG}$ revealed a free pool of mL62$^{FLAG}$ in fractions 1–3 (*Figure 1A*), a fraction of mL62$^{FLAG}$ that co-sedimented with the 39S mtLSU (fractions 7 and 8) and with the 55S mt-ribosome (fraction 10) as previously reported (*Richter et al., 2010*).

To identify mt-ribosome-associated proteins in human mitochondria, we defined the interactome of mL62$^{FLAG}$ by quantitative mass spectrometry using stable isotope labeling with amino acids in cell culture (SILAC) (*Figure 1B*, *Supplementary file 1*). As expected, we recovered all components of the mt-ribosome with high enrichment factors (*Figure 1B and C*). In addition, we found proteins of mtDNA and mtRNA maintenance, mt-ribosome biogenesis factors, and known OXPHOS assembly factors in the mL62$^{FLAG}$ interactome (*Figure 1C*). Additionally, we identified the uncharacterized protein TMEM223 and the putative mt-ribosome interacting protein SMIM4 (*Busch et al., 2019*). To confirm the interaction of TMEM223 and SMIM4 with the mt-ribosome, we performed immunoisolations of mL62$^{FLAG}$ from solubilized mitochondria and subjected the eluates to western blot analysis (*Figure 1D*). TMEM223 (*Figure 1D*, left panel) and SMIM4 (*Figure 1D*, right panel) were readily detectable in the eluate together with the ribosomal proteins uL1m and uS14m. Accordingly, TMEM223 and SMIM4 represent so far uncharacterized interactors of the mt-ribosome.

### TMEM223 is an inner mitochondrial membrane protein

The identification of the uncharacterized TMEM223 as a new mt-ribosome interacting protein led us to investigate the function of this protein. TMEM223 displays two putative transmembrane spans but lacks a defined N-terminal targeting sequence (*Figure 2A*). To address the submitochondrial localization of TMEM223, we subjected mitochondria to hypo-osmotic swelling and carbonate extraction experiments (*Figure 2B and C*). To detect TMEM223, we used an antibody directed against the C-terminus of the protein. Upon Proteinase K treatment of mitoplasts, a faster migrating C-terminal fragment of TMEM223 was detected (*Figure 2B*). This finding indicates that the C-terminus was exposed to the mitochondrial matrix and that the fragment represents the second transmembrane domain and the C-terminus. As TMEM223 was resistant to carbonate extraction (*Figure 2C*), we concluded that TMEM223 is an integral protein of the IMM with its N- and C-termini facing the mitochondrial matrix.

To investigate the function of TMEM223, we generated a TMEM223 knockout cell line (TMEM223$^{-/-}$) utilizing a CRISPR/Cas9 approach. For this, we targeted the TMEM223 gene (NM_001080501.3) and confirmed the genomic modification by sequencing. The TMEM223$^{-/-}$ cell line displayed multiple nucleotide exchanges that resulted in premature stop codons at codons encoding amino acids 36 and 38. Loss of TMEM223 was further confirmed by western blot analysis of purified mitochondria and the steady-state amounts of selected mitochondrial proteins investigated (*Figure 2D*). Quantifications

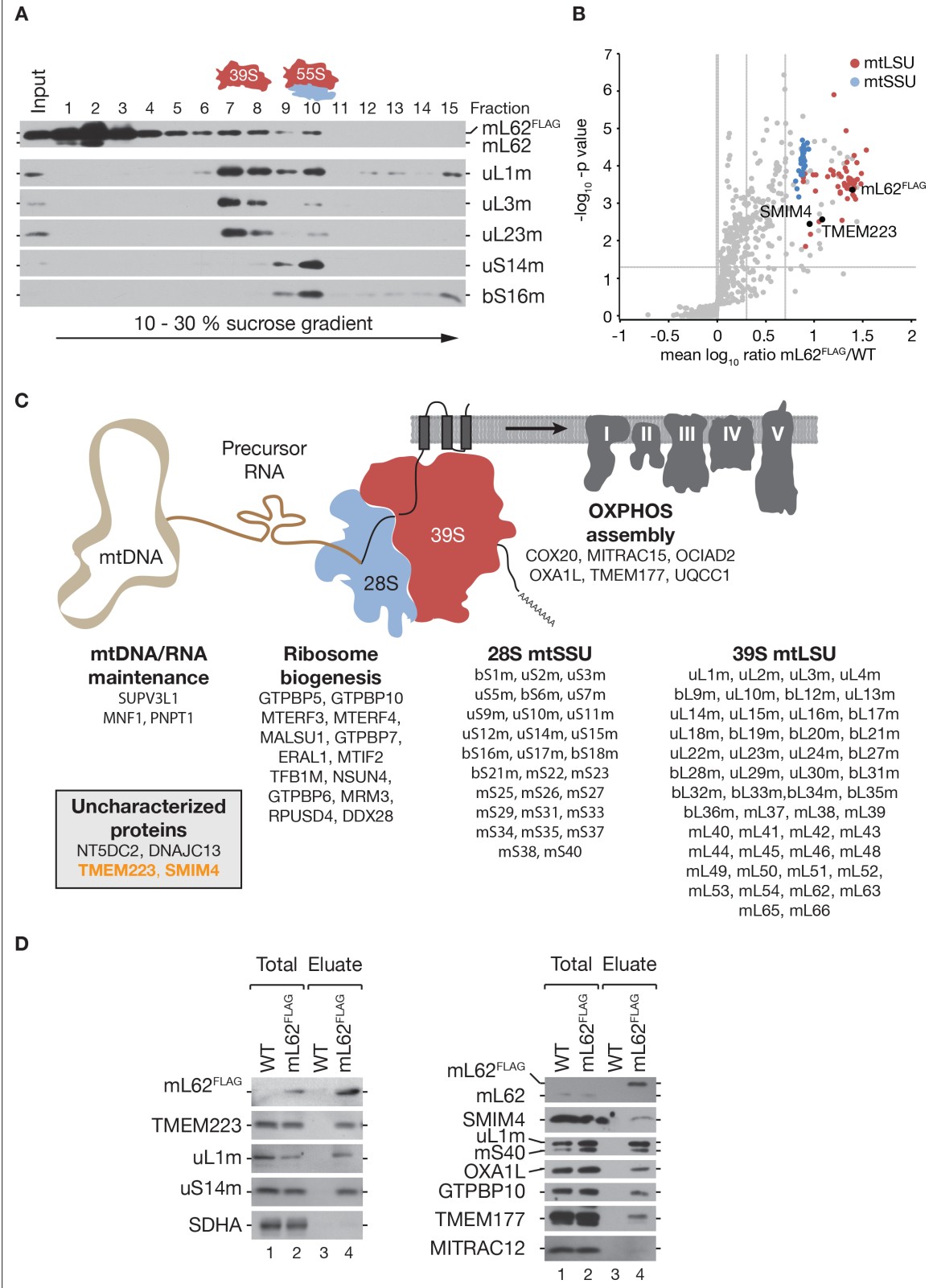

**Figure 1.** TMEM223 and SMIM4 interact with the mitochondrial ribosome. (**A**) Mitochondria isolated from cells expressing mL62^FLAG were subjected to co-immunoprecipitation. Natively isolated complexes were separated by sucrose density gradient ultracentrifugation. Fractions (1–15) were analyzed by western blotting, using indicated antibodies against components of the 39S mtLSU (mL62, uL1m, uL3m, and uL23m) and 28S mtSSU (uS14m and bS16m) subunits. (**B**) Mitochondria were isolated from wild-type (WT) and mL62^FLAG-expressing cells, cultured in SILAC-medium, and subjected to

*Figure 1 continued on next page*

*Figure 1 continued*

co-immunoisolation. Eluates were analyzed by quantitative mass spectrometry (LC-MS/MS) (n=4). Ribosomal proteins of the mtLSU and mtSSU are indicated in red and blue, respectively. Dashed lines indicate a p-value of 0.05 and mean mL62$^{FLAG}$/WT ratios. (C) Scheme of proteins identified in (B). (D) Complexes containing mL62$^{FLAG}$ were purified as in (A) and (B) and analyzed by western blotting (Total, 0.75%; Eluate, 100%).

The online version of this article includes the following source data for figure 1:

**Source data 1.** Data of *Figure 1A and D*.

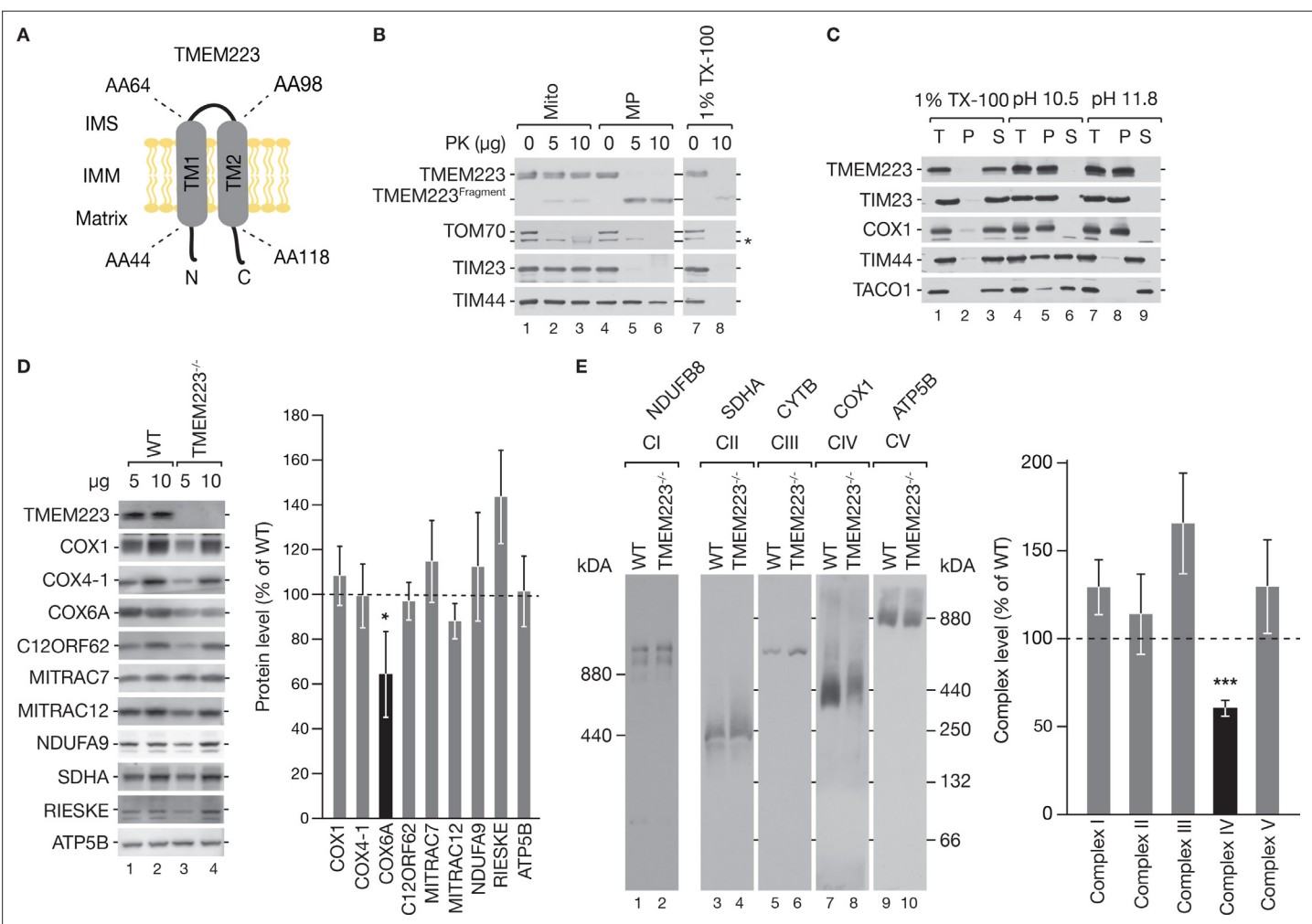

**Figure 2.** TMEM223 is a mitochondrial membrane protein. (A) Membrane topology of TMEM223. The predicted transmembrane spans (TM1 and TM2) with corresponding amino acids (aa) are indicated. IMS: intermembrane space; IMM: inner mitochondrial membrane. (B) and (C) Submitochondrial localization of TMEM223. Wild-type (WT) mitochondria were treated with Proteinase K (PK) under iso-osmotic (Mito), hyper-osmotic conditions (swelling, MP), or solubilized with Triton X-100 (TX-100) (B). The unspecific band is marked with an asterisk. Mitochondrial proteins were extracted in sodium carbonate containing buffer at different pH (total, T; pellet, P; soluble fraction, S) (C). (D) Protein steady-state levels in TMEM223$^{-/-}$ cells. Mitochondrial lysates from WT and TMEM223$^{-/-}$ cells were analyzed by western blotting using indicated antibodies and protein amounts were quantified using ImageQuant software (mean ± SEM, n=3). (E) Isolated mitochondria from WT and TMEM223$^{-/-}$ cells were solubilized in DDM-containing buffer, separated on 2.5–10% (Complex I) or 4–13% (Complexes II–V) BN-PAGE and analyzed by western blotting. OXPHOS complexes were detected with indicated antibodies and amounts quantified using ImageQuant software(mean ± SEM, n=3).

The online version of this article includes the following source data and figure supplement(s) for figure 2:

**Source data 1.** Data of *Figure 2B–E*.

**Figure supplement 1.** Loss of TMEM223 increases the level cytochrome *c* reductase.

**Figure supplement 1—source data 1.** Data of *Figure 2—figure supplement 1A, B, C*.

of the signals revealed a reduction of the late assembling complex IV subunit COX6A, while COX4-1 (an early complex IV constituent) or the COX1 assembly factors C12ORF62 (COX14) and MITRAC12 (COA3) were not altered (*Figure 2D*). Also, proteins of other OXPHOS complexes, such as NDUFA9 (complex I), SDHA (complex II), or ATP5B (complex V), remained unaffected. Interestingly, we observed an increase of RIESKE, a core protein of complex III (*Figure 2D*).

The reduction of COX6A prompted us to investigate mitochondrial OXPHOS complexes by Blue Native (BN)-PAGE (equal loading was confirmed by SDS-PAGE (*Figure 2—figure supplement 1A*)). Using n-Dodecyl-β-D-maltosid (DDM) for mitochondrial solubilization, we observed a selective reduction of the cytochrome *c* oxidase to 60% of the WT control (*Figure 2E*). However, while complexes I, II, and IV were only marginally more abundant, complex III showed an increase (approx. 165% of WT) in the TMEM223$^{-/-}$ cell line (*Figure 2E*) independent of the detergent used for solubilization (similar results were obtained using Digitonin; *Figure 2—figure supplement 1B*). To better understand the increase of complex III, we analyzed the steady-state levels of complex III associated proteins (*Figure 2—figure supplement 1C*) as well as the stability of the core subunit RIESKE and of the assembly factors UQCC1 and C12ORF73 (*Figure 2—figure supplement 1D*, Figure 5 and 6). For stability analysis, the amount of these proteins was determined 48 hr after inhibition of mitochondrial translation by thiamphenicol, since the synthesis of the mitochondrial-encoded complex III protein CYTB is required for RIESKE stability (*Protasoni et al., 2020*). Although we observed an increase in the steady-state levels of RIESKE (*Figure 2D*), UQCC2, and C12ORF73 in the absence of TMEM223 (*Figure 2—figure supplement 1C*), we did not detect a difference in stability of the tested proteins under these conditions (*Figure 2—figure supplement 1D*). We concluded that loss of TMEM223 leads to a reduction of cytochrome *c* oxidase.

## TMEM223 is involved in cytochrome *c* oxidase biogenesis

For a quantitative assessment, we measured cytochrome *c* oxidase activity using a colorimetric assay. In agreement with the BN-PAGE analyses (*Figure 2E*), the activity of the cytochrome *c* oxidase in TMEM223$^{-/-}$ cells was reduced to 62.5% of wild-type (WT) (*Figure 3A*).

Since we identified TMEM223 as an mt-ribosome-associated protein and observed a reduction of the cytochrome *c* oxidase, we asked whether TMEM223 was required for translation of mitochondrial-encoded complex IV subunits. Therefore, we performed [$^{35}$S]methionine labeling of mitochondrial translation products. These analyses revealed a significant decrease in the levels of newly synthesized COX1 in TMEM223$^{-/-}$ cells compared to WT while other mitochondrial-encoded proteins, including COX2 and COX3, displayed no differences (*Figure 3B*). We confirmed the reduced COX1 synthesis rate in the TMEM223$^{-/-}$ in a complementary approach. Using a [$^{35}$S]methionine labeling approach in siRNA-mediated depleted TMEM223 cells, we observed a comparable reduction of COX1 synthesis to 61.23% (*Figure 3C*).

Our data suggested an involvement of TMEM223 in the early stages of cytochrome *c* oxidase biogenesis. To address this, we performed immunoisolations using tagged constituents of the early COX1 assembly intermediates (MITRAC complexes), namely C12ORF62 (COX14) (*Richter-Dennerlein et al., 2016*; *Weraarpachai et al., 2012*; *Figure 3D*), MITRAC12 (COA3) (*Mick et al., 2012*; *Figure 3E*), and CMC1 (*Horn et al., 2008*; *Mick et al., 2012*; *Timón-Gómez et al., 2018*; *Figure 3F*). As a control, we used MITRAC7, a later stage assembly factor of COX1 (*Dennerlein et al., 2015*; *Figure 3G*). TMEM223 was recovered in the eluate of C12ORF62$^{FLAG}$ and MITRAC12$^{FLAG}$, but was not present in CMC1$^{FLAG}$ purifications and we detected only marginal amounts in the eluate of MITRAC7$^{FLAG}$ (*Figure 3D–G*). These findings support the role of TMEM223 in the first steps of cytochrome *c* oxidase biogenesis and show that it interacts with early MITRAC complexes.

## Protein interaction network of SMIM4

Our mass spectrometric analysis identified the uncharacterized protein SMIM4 as a new mt-ribosome-associated protein (*Figure 1B and D*). SMIM4 displays one predicted transmembrane domain (amino acids 20–41) (*Figure 4A*). To investigate the localization and function of SMIM4, we generated a stable HEK293T cell line, allowing for inducible expression of a C-terminally FLAG-tagged variant of SMIM4 (SMIM4$^{FLAG}$). We investigated the subcellular localization of SMIM4$^{FLAG}$ by STED super-resolution light microscopy using a FLAG-specific antibody. Comparison with a TOM22 specific antibody labeling showed that SMIM4$^{FLAG}$ localizes to mitochondria (*Figure 4B*). To investigate the submitochondrial

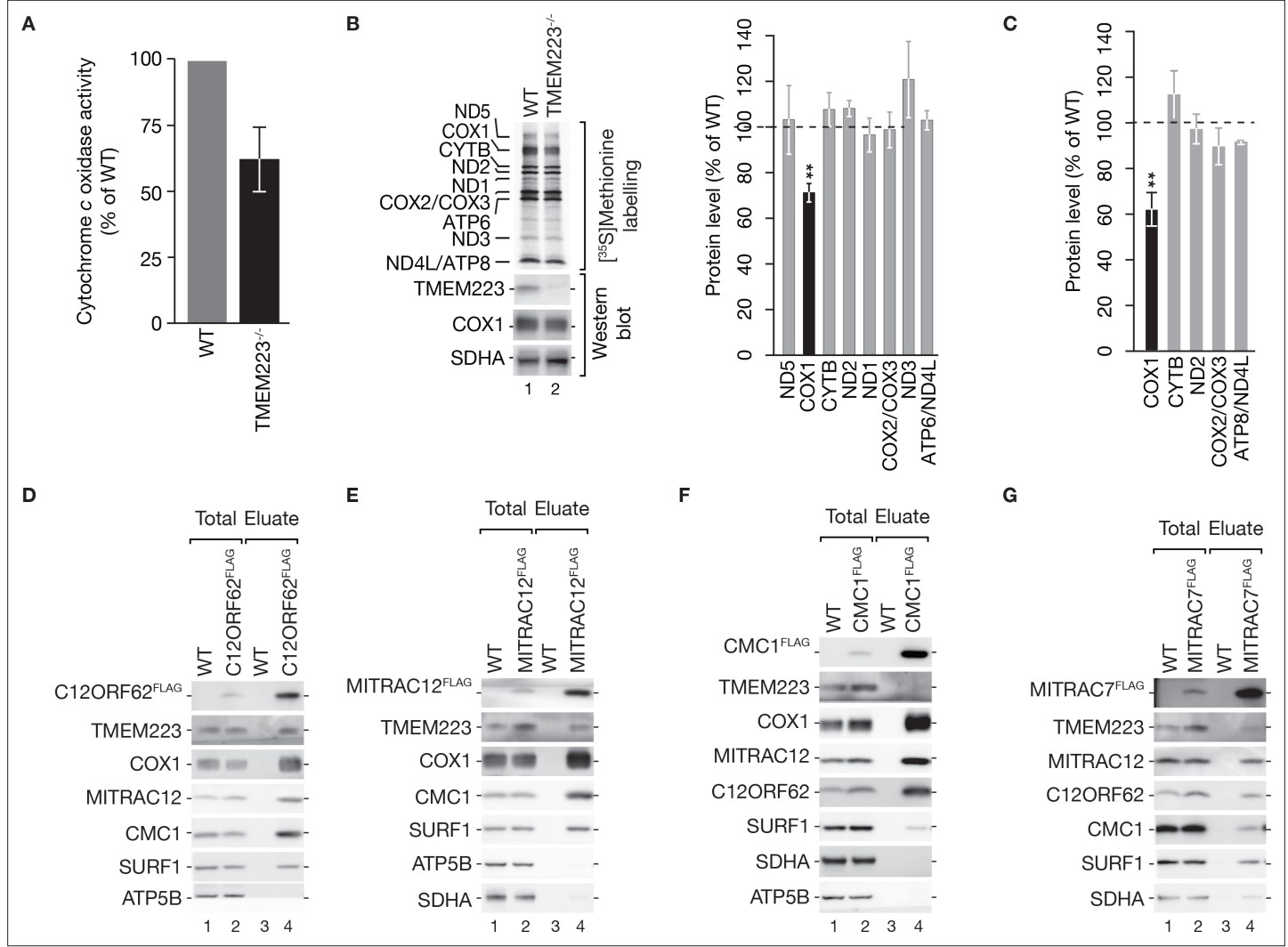

**Figure 3.** TMEM223 is involved in cytochrome *c* oxidase biogenesis. (**A**) Cytochrome *c* oxidase activity was measured in cellular extracts from wild-type (WT) and TMEM223$^{-/-}$ photometrically (mean ± SEM, n=4). (**B, C**) Mitochondrial protein synthesis in TMEM223$^{-/-}$ (**B**) or siRNA mediated TMEM223 depleted (**C**) cells. Cells were grown in the presence of [$^{35}$S]methionine for 1 hr to monitor synthesis of mitochondrial-encoded proteins. Cell lysates were subjected to SDS-PAGE and analyzed by digital autoradiography and western blotting (lower panel in (**B**)). Newly synthesized mitochondrial proteins were quantified, using the ImageQuant software, calculated as percentage of WT and internally standardized to ATP6 (mean ± SEM; n=3). (**D**–**G**) TMEM223 interacts with early cytochrome *c* oxidase assembly factors. Mitochondria isolated from WT, C12ORF62$^{FLAG}$ (**D**), MITRAC12$^{FLAG}$ (**E**), CMC1 (**F**), or MITRAC7$^{FLAG}$ (**G**) cells were subjected to co-immunoisolation and samples analyzed by western blotting (Total, 0.75%; Eluate, 100%).

The online version of this article includes the following source data for figure 3:

**Source data 1.** Data of *Figure 3B, D, E, F, G*.

localization of SMIM4, we performed hypo-osmotic swelling and carbonate extraction experiments. The C-terminal FLAG epitope of SMIM4 became accessible to Proteinase K digestion upon disruption of the outer mitochondrial membrane (*Figure 4C*). Furthermore, SMIM4 was resistant to carbonate extraction (*Figure 4D*) indicating that SMIM4 is an integral IMM protein facing its C-terminus to the intermembrane space (IMS).

To assess the function of SMIM4, we first defined its interactome. For this, we analyzed SMIM4$^{FLAG}$-containing protein complexes, purified from mitochondria following a SILAC-based quantitative mass spectrometry approach (*Figure 4E*, *Supplementary file 2*). SMIM4$^{FLAG}$ efficiently isolated components of the 28S mtSSU and 39S mtLSU supporting its interaction with the mt-ribosome (*Figure 1C and D*). Additionally, we identified in SMIM4$^{FLAG}$ complexes the cytochrome *c* reductase (complex III) assembly factors UQCC1, UQCC2, and C12ORF73 (*Zhang et al., 2020*; *Tucker et al., 2013*; *Ndi et al., 2018*;

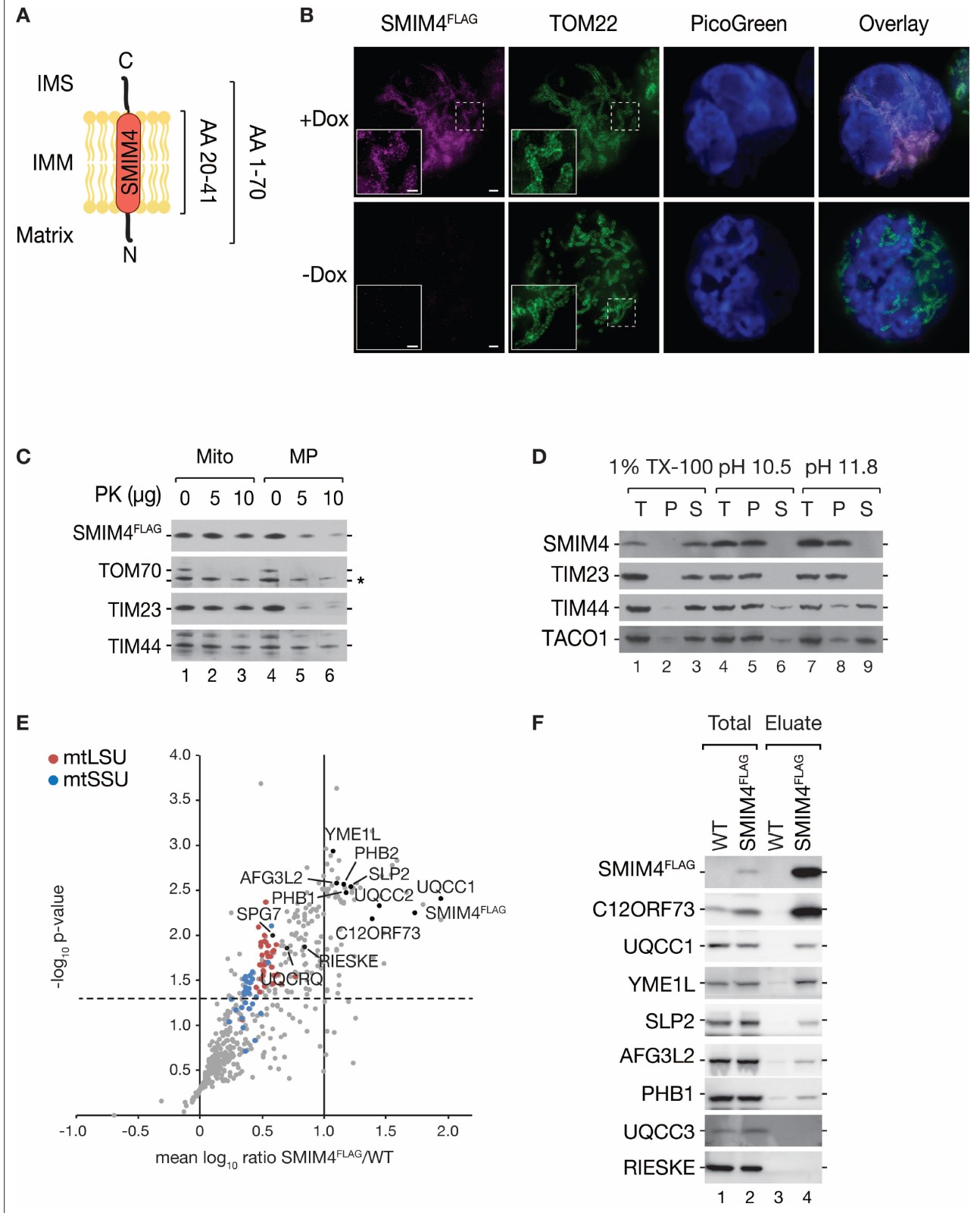

**Figure 4.** SMIM4 is a mitochondrial protein, interacting with cytochrome *c* reductase and mitochondrial quality control proteins. (**A**) Schematic presentation of SMIM4 membrane topology. The predicted transmembrane domain (AA 20–41) is indicated. IMS: intermembrane space; IMM, inner mitochondrial membrane. (**B**) Immunofluorescence microscopy of HEK293T cells expressing SMIM4^FLAG. Cells were induced with doxycycline for 24 hr (+Dox). SMIM4^FLAG was labeled using a FLAG-specific antiserum. As a mitochondrial marker, a TOM22 specific antiserum was used. SMIM4^FLAG and

*Figure 4 continued on next page*

Figure 4 continued

TOM22 were imaged with STED microscopy, PicoGreen by confocal microscopy. DNA was labeled via Quant-iT PicoGreen dsDNA reagent. Scale bars: 10 μm (overview), 500 nm (magnification). (**C**) Submitochondrial localization of SMIM4$^{FLAG}$. Mitochondria isolated from SMIM4$^{FLAG}$-expressing cells were treated with proteinase K (PK) either under iso-osmotic (Mito), hyperosmotic (swelling, mitoplasts [MP]) conditions. Unspecific band is marked with an asterisk. (**D**) SMIM4 is an integral membrane protein. Mitochondrial proteins were extracted using sodium carbonate (at indicated pH), or Triton X-100 (TX-100). Samples (total, T; pellet, P; soluble fraction, S) were analyzed by western blotting using antibodies. (**E**) Mitochondrial extracts from wild-type (WT) and SMIM4$^{FLAG}$-expressing cells cultured in SILAC medium were subjected to native-immunoprecipitation, and analyzed by quantitative mass spectrometry (LC-MS/MS) (n=4). Cytochrome $c$ reductase assembly factors and components of the mitochondrial quality control system are indicated in black. The dashed horizontal line indicates a p-value of 0.05, the solid vertical line a mean SMIM4$^{FLAG}$/WT ratio of 10. (**F**) Samples obtained by co-immunoprecipitation of WT or SMIM4$^{FLAG}$-containing mitochondrial lysates were analyzed by western blotting (Total, 1.5%; Eluate, 100 %).

The online version of this article includes the following source data for figure 4:

**Source data 1.** Data of *Figure 4C, D, F*.

*Fernandez-Vizarra and Zeviani, 2018*) as well as components of the mitochondrial quality control system, such as the *m*AAA-protease (AFG3L2 and SPG7), the *i*AAA-protease (YME1L), the membrane scaffolds SLP2 and prohibitins (PHB1 and PHB2). In agreement with the mass spectrometric analysis, we detected YME1L, SLP2, AFG3L2 and PHB1, C12ORF73 and UQCC1 in the SMIM4$^{FLAG}$ eluate by immunoblotting (*Figure 4F*). Interestingly, proteins acting in later assembly steps of the cytochrome $c$ reductase such as UQCC3 (not detected) or RIESKE and UQCRQ (*Wanschers et al., 2014*; *Ndi et al., 2018*; *Fernandez-Vizarra and Zeviani, 2018*) were less enriched in the SILAC analysis (*Figure 4E*). Furthermore, we could not detect UQCC3 or RIESKE by western blot (*Figure 4F*). In summary, SMIM4 is an integral protein of the IMM that interacts with the mitochondrial ribosome, the mitochondrial quality control machinery and preferentially early cytochrome $c$ reductase biogenesis factors.

## C12ORF73 and SMIM4 are involved in cytochrome $c$ reductase maturation

C12ORF73 was recently found to contribute to cytochrome $c$ reductase biogenesis. However, its function in the assembly process remains elusive. To define its role in cytochrome $c$ reductase assembly, we generated an inducible C12ORF73$^{FLAG}$-expressing HEK293T cell line. We confirmed mitochondrial localization of C12ORF73$^{FLAG}$ by STED super-resolution light microscopy, using antisera specific for the FLAG-tag and the outer mitochondrial membrane protein TOM22, respectively. (*Figure 5— figure supplement 1A*). The C12ORF73$^{FLAG}$ signal was superimposable with the mitochondrial marker TOM22 supporting mitochondrial localization of the protein in human cells. Biochemical analyses to determine the submitochondrial localization of C12ORF73$^{FLAG}$ showed that the protein was resistant to carbonate extraction and that its C-terminus was accessible to protease treatment when the outer membrane was disrupted by hypo-osmotic swelling (*Figure 5—figure supplement 1B and C*). These results confirmed that C12ORF73 represents a protein of the IMM that exposes its C-terminus into the IMS, which agrees with previous studies (*Zhang et al., 2020*).

Since we found C12ORF73 together with early cytochrome $c$ reductase assembly factors in the interactome of SMIM4$^{FLAG}$ (*Figure 4E and F*), we dissected the interaction of C12ORF73 with the cytochrome $c$ reductase assembly machinery. To this end, we subjected mitochondrial extracts of C12ORF73$^{FLAG}$-containing mitochondria to immunoisolations and analyzed the eluates by western blotting (*Figure 5A*). Similar to SMIM4 co-immunoisolations (*Figure 4*), we recovered UQCC1 and UQCC2, but not UQCC3 or RIESKE, co-isolating with C12ORF73$^{FLAG}$ (*Figure 5A*). *Zhang et al., 2020* reported an interaction of C12ORF73 (BRAWNIN) with UQCRQ using transient co-transfection of C12ORF73$^{HA}$ and UQCRQ$^{FLAG}$. We repeated the transient transfection of UQCRQ$^{FLAG}$ into HEK293T cells. Although we could isolate UQCRC2 and SMIM4, only marginal amounts of UQCC1 or C12ORF73 were co-isolated (*Figure 5—figure supplement 1D*). Hence, to support our conclusion that SMIM4 and C12ORF73 act early in cytochrome $c$ reductase assembly, we performed immunoisolations using an antibody against the RIESKE protein. While we isolated UQCRC1 and UQCC3, the early cytochrome $c$ reductase assembly factors UQCC1, C12ORF73, and SMIM4 were not co-isolated (*Figure 5—figure supplement 1E*). This analysis further supports the early role of C12ORF73 and SMIM4 in the biogenesis pathway of the cytochrome $c$ reductase.

As SMIM4 (*Figure 4E and F*) and C12ORF73 (*Figure 5A*) isolated early cytochrome $c$ reductase assembly factors, we determined the function of these proteins utilizing siRNA applications. Hence,

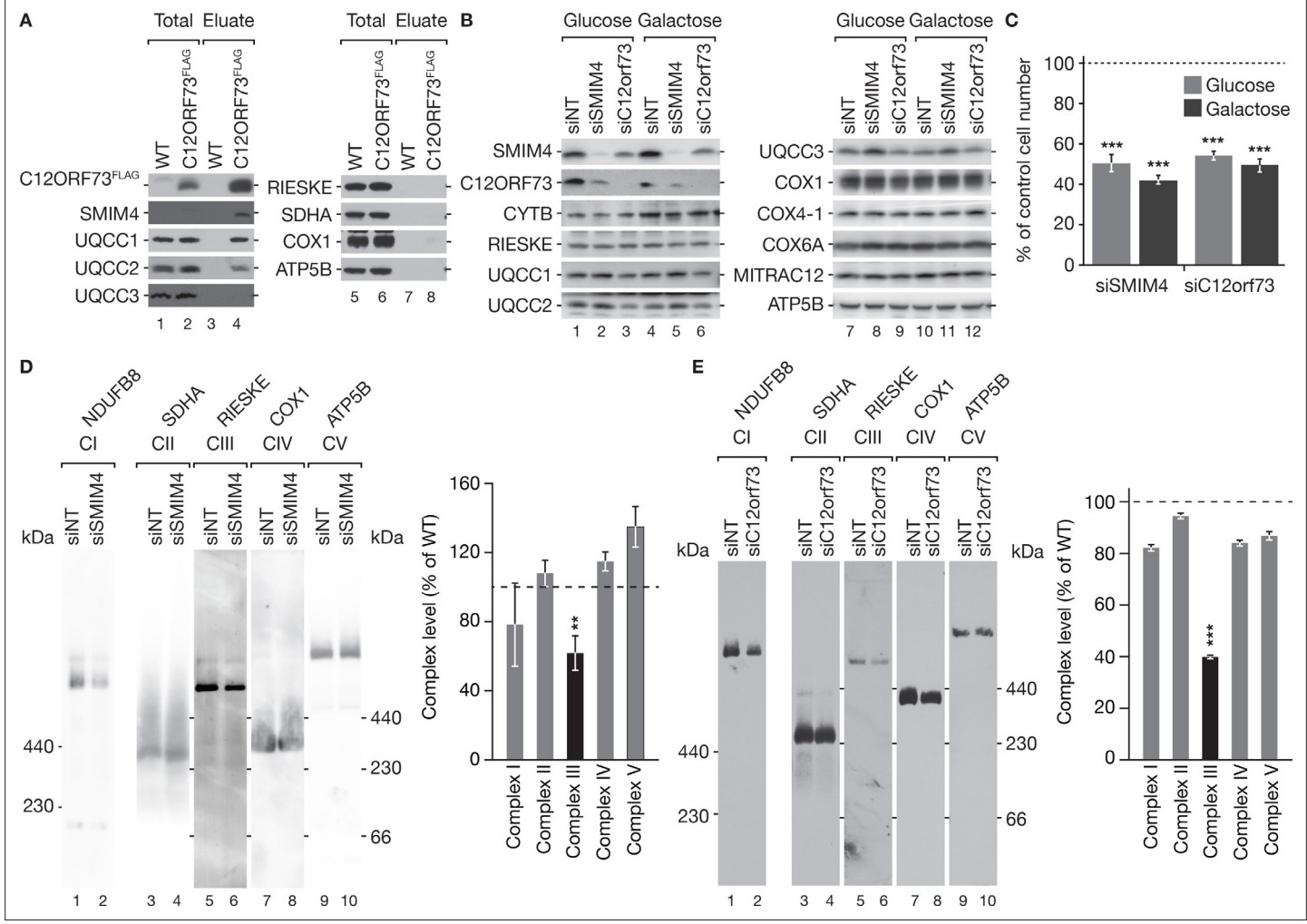

**Figure 5.** Loss of SMIM4 or C12ORF73 affects cytochrome *c* reductase biogenesis. (**A**) FLAG-immunoisolation of C12ORF73[FLAG]. Samples were analyzed by SDS-PAGE and western blotting (Total, 1.5%; Eluate, 100 %). (**B**) Western blot analyses of SMIM4 or C12ORF73 depleted cells. HEK293T cells were treated with indicated siRNAs and cultured either in glucose- or galactose-containing media for 72 hr. Cell extracts were subjected to SDS-PAGE separation and western blotting. (**C**) Loss of SMIM4 or C12ORF73 affects cell growth. HEK293T cells were transfected with siRNAs as in *Figure 6B*. Cells were cultured either in glucose- or galactose-containing media for 72 hr; cell counts are presented as percentage relative to non-targeting siRNA-treated cells (siNT; indicated as dashed line) (mean ± SEM, n=3). (**D, E**) BN-PAGE analyses of mitochondrial protein complexes upon SMIM4 (**B**) or C12ORF73 ablation (**C**). Mitochondria were solubilized with DDM (N-Dodecyl-beta-Maltoside) and subjected to BN-PAGE followed by western blot analyses. OXPHOS complex levels were quantified using the ImageQuant software and graphed (mean ± SEM, n=3).

The online version of this article includes the following source data and figure supplement(s) for figure 5:

**Source data 1.** Data of *Figure 5A,B, D, E*.

**Figure supplement 1.** C12ORF73 is an inner mitochondrial protein in human.

**Figure supplement 1—source data 1.** Data of *Figure 5—figure supplement 1B–D*.

**Figure supplement 1—source data 2.** Data of *Figure 5—figure supplement 1E–G*.

we investigated protein levels in whole-cell extracts after siRNA-mediated depletion. SMIM4 was reduced in C12ORF73-depleted cells and vice versa SMIM4 ablation led to decreased C12ORF73 levels indicating an interdependency of these factors (*Figure 5B*). While we observed a subtle increase of UQCC3 in SMIM4-depleted cells, UQCC1 and UQCC2 were slightly reduced upon C12ORF73 knockdown (*Figure 5B*). To investigate cell viability upon SMIM4 or C12ORF73 depletion, we quantified cell numbers after 72 hr siRNA-mediated knockdown in glucose- or galactose-containing media

(*Figure 5C*). In both cases, we observed a drastic reduction in cell growth to approximately 50% compared to the siNT-control treated cells.

Considering the interaction of SMIM4 (*Figure 4E and F*) and C12ORF73 (*Figure 5E*) with cytochrome *c* reductase assembly factors and the growth phenotype upon 72 hr protein depletion (*Figure 5C*), we assessed mitochondrial OXPHOS complexes upon SMIM4 or C12ORF73 depletion by BN-PAGE analyses (equal loading was confirmed by SDS-PAGE) (*Figure 5D and E*, *Figure 5—figure supplement 1F* and G). While siRNA-mediated knockdown of SMIM4 led to a subtle increase of the ATP synthase, a significant reduction of the cytochrome *c* reductase to 62%, and to a minor but statistically not significant reduction of the NADH:ubiquinone oxidoreductase. However, other OXPHOS complexes remained unaffected (*Figure 5D*). Knockdown of C12ORF73 impaired slightly the level of

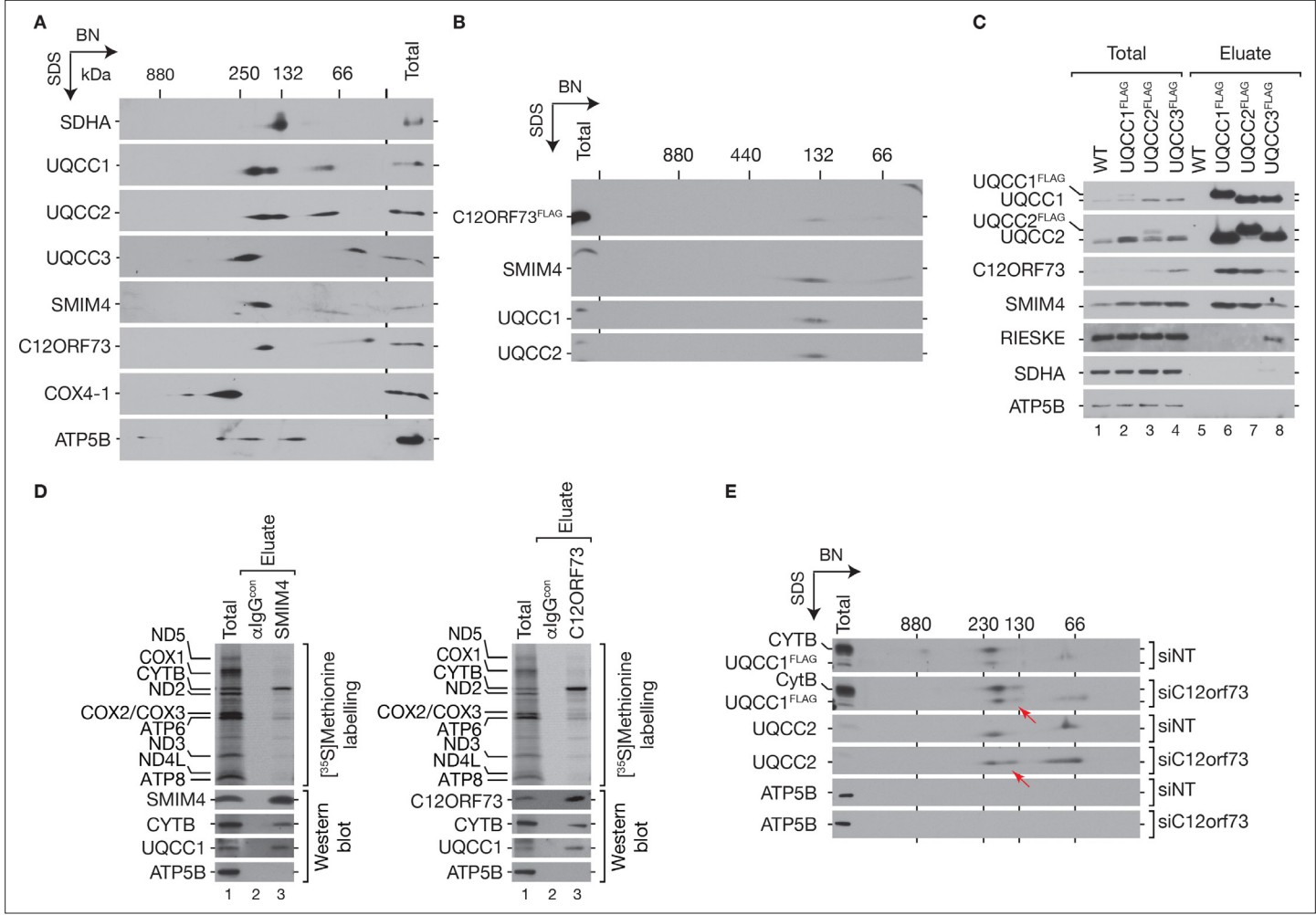

**Figure 6.** SMIM4 and C12ORF73 promote cytochrome *c* reductase assembly. (**A**) SMIM4 and C12ORF73 co-migrate with cytochrome *c* reductase assembly intermediates. Wild-type mitochondrial lysates were subjected to BN-PAGE, followed by second dimension SDS-PAGE and western blotting. (**B**) C12ORF73<sup>FLAG</sup> isolates cytochrome *c* reductase assembly intermediates. Mitochondria isolated from C12ORF73<sup>FLAG</sup>-expressing cells were solubilized and subjected to co-immunoisolation. Natively eluted complexes were separated by BN-PAGE and subjected to second dimension SDS-PAGE followed by western blot analyses using indicated antibodies. (**C**) Immunoprecipitations of UQCC1<sup>FLAG</sup>, UQCC2<sup>FLAG</sup>, and UQCC3<sup>FLAG</sup>. Eluates were analyzed by SDS-PAGE followed by western blotting with indicated antibodies. (**D**) Mitochondrial translation products were labeled with [$^{35}$S]methionine for 1 hr prior to co-immunoprecipitation using anti-SMIM4, -C12ORF73, or control antibodies (αIgG<sup>con</sup>). Eluates were separated by SDS-PAGE followed by western blotting, and analyzed by digital autoradiography (upper panel) and immunodetection (lower panel) (Total, 2%; Eluate, 100%). (**E**) Mitochondria isolated from control or C12ORF73-depleted cells were lysed in digitonin-containing buffer and complexes separated by BN-PAGE followed by second dimension SDS-PAGE. Cytochrome *c* reductase sub-assembly complexes were monitored with indicated antibodies (red arrow mark cytochrome *c* reductase subcomplexes in C12ORF73-deficient samples).

The online version of this article includes the following source data for figure 6:

**Source data 1.** Data of *Figure 6A-E*.

the NADH:ubiquinone oxidoreductase, cytochrome *c* oxidase, and the ATP synthase; however, the cytochrome *c* reductase was, similar to SMIM4 knockdown, significantly reduced (*Figure 5E*).

## SMIM4 and C12ORF73 participate in early steps of cytochrome c reductase assembly

To define SMIM4- and C12ORF73-containing complexes in cytochrome *c* reductase biogenesis, we solubilized mitochondria under mild conditions and separated protein complexes by first dimension BN-PAGE followed by a second dimension SDS-PAGE (*Figure 6A*). UQCC1 and UQCC2 were detected in two complexes between 132 and 250 kDa and in a smaller complex of approximately 70 kDa. UQCC3 appeared in a complex that migrated slightly slower than the biggest complex of UQCC1 and UQCC2 as well as in a complex below 66 kDa. SMIM4- and C12ORF73-containing complexes mainly co-migrated with the largest UQCC1- and UQCC2-containing complex, but moderately faster than the biggest UQCC3 complex. These data supported the idea that SMIM4 and C12ORF73 are part of an early cytochrome *c* reductase assembly intermediate with UQCC1 and UQCC2. When we isolated C12ORF73[FLAG]-containing complexes from mitochondria and subjected these purified complexes to BN-PAGE followed by SDS-PAGE, C12ORF73[FLAG] isolated UQCC1, UQCC2, and SMIM4, all of which migrated in a single complex (*Figure 6B*). These results demonstrate that C12ORF73[FLAG] forms a complex with UQCC1, UQCC2, and SMIM4 but not with UQCC3. In agreement with this, SMIM4 and C12ORF73 were efficiently isolated with UQCC1[FLAG] and UQCC2[FLAG] but only in minor amounts with UQCC3[FLAG]. RIESKE was only present in the UQCC3[FLAG] eluate (*Figure 6C*).

The interaction between SMIM4, C12ORF73, UQCC1, and UQCC2 as early cytochrome *c* reductase assembly factors implies that SMIM4 and C12ORF73 interact with newly synthesized cytochrome *b* (CYTB). Therefore, we performed [35S]methionine labeling of newly translated mitochondrial-encoded proteins and subjected cellular extracts to immunoisolation using antibodies against SMIM4 or C12ORF73 (*Figure 6D*). As expected, both immunoisolations clearly enriched newly synthesized CYTB.

As SMIM4 and C12ORF73 were present in early assembly intermediates and their loss led to a reduction of the cytochrome *c* reductase, we addressed at which stage a loss of C12ORF73 affects cytochrome *c* reductase assembly. For this, we determined assembly intermediates upon loss of C12ORF73. To this end, we depleted C12ORF73 in UQCC1[FLAG]-expressing cells and subjected mitochondrial extracts to immunoisolations. Purified complexes were analyzed by BN-PAGE followed by SDS-PAGE (*Figure 6E*). In comparison to the control, we observed an accumulation of a cytochrome *c* reductase subcomplex containing CYTB, UQCC1[FLAG], and UQCC2 in C12ORF73-ablated cells (*Figure 6E*, red arrow). In summary, our work defined SMIM4 and C12ORF73 as two new factors that act early in the biogenesis of the cytochrome *c* reductase.

## Discussion

Recent analyses shed light on the mitochondrial translation machinery and provide an understanding of the biogenesis and the structural organization of the mt-ribosome (*Kummer and Ban, 2021*; *Amunts et al., 2015*; *Brown et al., 2014*; *Hällberg and Larsson, 2014*; *Maiti et al., 2021*; *Ferrari et al., 2021*). Additionally, factors that regulate the translation of mitochondrial-encoded proteins have been identified (*Kummer and Ban, 2021*). However, the mechanisms underlying coordination of mitochondrial translation with OXPHOS assembly and the involved factors remain poorly understood (*Richter-Dennerlein et al., 2015*; *Dennerlein et al., 2017*; *Kummer and Ban, 2021*; *Hällberg and Larsson, 2014*). Especially, the question as to how long the mitochondrial ribosome remains associated with biogenesis intermediates of the oxidative phosphorylation system is still unaddressed (*Richter-Dennerlein et al., 2016*; *Maiti et al., 2021*; *Ferrari et al., 2021*; *Lavdovskaia et al., 2021*).

In this study, we aimed to define the interactome of the mt-ribosome in human cells with a special focus on its membrane-bound interaction partners. Using mild solubilization and purification conditions, we identified previously described auxiliary factors (*Busch et al., 2019*; *Kummer and Ban, 2021*) and the IMM proteins TMEM223 and SMIM4 as mt-ribosome interactors. Furthermore, we identified NT5DC2 and DNAJC13. The molecular function of these proteins is largely unknown and if they could act as OXPHOS assembly factors requires further investigation. Interestingly, we also identified assembly factors of respiratory chain complexes IV, such as COX20 (*Bourens et al., 2014*; *Lorenzi*

*et al., 2018*) or COA1 (*Wang et al., 2020*). These findings suggest that the mitochondrial ribosome and its interacting partners could act as a platform to initiate OXPHOS assembly and that the mitochondrial ribosome remains associated with maturing complexes beyond the cotranslational assembly stages (*Richter-Dennerlein et al., 2016*; *Maiti et al., 2021*; *Ferrari et al., 2021*; *Lavdovskaia et al., 2021*; *Mick et al., 2012*).

However, in agreement, a recent complementary proteomic study in mouse, which identified new mt-ribosome biogenesis factors, also listed SMIM4 in the group of uncharacterized mt-ribosome interacting proteins (*Busch et al., 2019*). We show that TMEM223 is involved in assembly of the cytochrome *c* oxidase, while SMIM4 and the interacting C12ORF73 contribute to the biogenesis of the cytochrome *c* reductase.

We show that TMEM223 is an IMM protein. Thus, our data represent experimental evidence for a previous prediction (*Sánchez-Caballero et al., 2020*). TMEM223 acts early in the assembly process of the mitochondrial-encoded COX1 and is required for cytochrome *c* oxidase assembly. Concomitantly, the cytochrome *c* reductase increases in the TMEM223 knockout cell line, potentially as a stress response to compensate the reduction of the cytochrome *c* oxidase. Interestingly, depletion of TMEM223 affects COX1 synthesis. Although TMEM223 is lacking in the knockout cell line mitochondria maintain reduced amounts of cytochrome *c* oxidase. It is tempting to speculate that the function of TMEM223 is partially overlapping with other assembly factors, such as C12ORF62 (COX14) or MITRAC12 (COA3) or that TMEM223 acts as a chaperone for cytochrome *c* oxidase assembly, rather than presenting an essential assembly factor. However, these hypotheses need to be tested further. The reduction in cytochrome *c* oxidase is in line with the observation that TMEM223 is predominantly found in the early assembly MITRAC intermediate together with C12ORF62 (COX14). C12ORF62 (COX14) was previously shown to bind mt-ribosomes and is required for efficient COX1 translation (*Richter-Dennerlein et al., 2016*; *Weraarpachai et al., 2012*; *Richter-Dennerlein et al., 2015*; *Dennerlein et al., 2017*). However, TMEM223 is released from COX1 during assembly of other subunits, as it is only marginally associated with later MITRAC complexes, which are characterized by the presence of MITRAC7. These findings support the idea of TMEM223 representing a new early COX1 assembly factor that links biogenesis processes with translation.

In addition to TMEM223, we identified SMIM4 as an mt-ribosome-associated factor. Our work links SMIM4 to known early assembly factors for complex III (UQCC1 and UQCC2), and we identified C12ORF73 as an SMIM4-associated protein. Both, SMIM4 and C12ORF73, are functional interdependent, since loss of C12ORF73 leads to a reduction of SMIM4 and vice versa loss of SMIM4 leads to a reduction of C12ORF73. We show that SMIM4 and C12ORF73 regulate cytochrome *c* reductase assembly. However, cells depleted for SMIM4 or C12ORF73 maintain reduced amounts of the cytochrome *c* reductase. Considering the knockdown conditions used in our study, it is feasible that the remaining low amounts of SMIM4 or C12ORF73 can still participate in the biogenesis process. Another consideration would be that the half-life of the cytochrome *c* reductase could exceed the time frame of the siRNA application. *Zhang et al., 2020* reported a strong reduction, but not a loss, of the cytochrome *c* reductase in model organisms. They pointed out that C12ORF73 (BRAWNIN) could also be involved in the import of cytochrome *c* reductase subunits or that it is involved in sensing nutrient availability and the energy status of the cell (*Zhang et al., 2020*). However, to dissect these possibilities will require further investigations. The interactions with newly synthesized CYTB, UQCC1, and UQCC2 position both proteins in the very early steps of the assembly process. Assembly of the cytochrome *c* reductase has been poorly investigated in mammals but is considered to recapitulate the process in the yeast system based on the high similarity in structure and composition of the yeast and human complexes (*Zara et al., 2009*; *Smith et al., 2012*; *Fernandez-Vizarra and Zeviani, 2018*; *Ndi et al., 2018*). The assembly process initiates with the synthesis of CytB and the association of the translational regulators Cbp3 (human UQCC1) and Cbp6 (UQCC2). Subsequently, Cbp3 (UQCC3) joins CytB. SMIM4 and C12ORF73 interact predominantly with UQCC1 and UQCC2 but not with UQCC3. Accordingly, both proteins are involved in the early steps of cytochrome *c* reductase assembly. Interestingly, SMIM4 and C12ORF73 are not conserved in yeast, indicating differences in cytochrome *c* reductase assembly between yeast and human. A recent study (*Stephan and Ott, 2020*) investigated the dimerization of complex III. The authors showed that complex III dimerization takes place early in the assembly process, when Cyt*b* maturation is completed and the protein is fully hemylated. However, if SMIM4 or C12ORF73 are required for dimerization processes within cytochrome *c*

reductase assembly processes would require further investigation. In a recent analysis, the zebrafish homolog of C12ORF73, BRAWNIN (BR), was suggested to contribute to cytochrome *c* reductase assembly (*Zhang et al., 2020*). However, depletion experiments did not reveal any assembly intermediates. Accordingly, BR was suggested to act prior to complex dimerization. On the other hand, BN- and SDS-PAGE analyses indicated that BR partially co-migrated with the dimerized form of the cytochrome *c* reductase (*Zhang et al., 2020*). In our study, siRNA-mediated depletion of C12ORF73 or SMIM4 caused a reduction of the cytochrome *c* reductase. In contrast to the reports on BR (*Zhang et al., 2020*), C12ORF73 does not interact with late complex constituents but rather shows exclusively an association with the early assembly factors UQCC1 and UQCC2. This observation positions SMIM4 and C12ORF73 to early assembly steps rather than to the late complex formation stages. This is further underlined by the fact that both proteins are only present in the UQCC1 and UQCC2 assembly intermediates.

In conclusion, we defined the interactome of the human mt-ribosome and identified three previously uncharacterized IMM proteins as mt-ribosome-associated factors. Importantly, we unravel their functions as cytochrome *c* reductase and cytochrome *c* oxidase assembly factors that support the immediate assembly of newly synthesized proteins into further assembly intermediates. Functional characterization defined TMEM223 as essential for cytochrome *c* oxidase maturation, while SMIM4/C12ORF73 acting as cytochrome *c* reductase biogenesis factors.

# Materials and methods

**Key resources table**

| Reagent type (species) or resource | Designation | Source or reference | Identifiers | Additional information |
|---|---|---|---|---|
| Cell line (*Homo sapiens*) | HEK293-Flp-InTM T-RexTM (HEK293T) Cell Line | Thermo Fisher Scientific | RRID:CVCL_U421 | |
| Cell line (*H. sapiens*) | HEK293-Flp-InTM T-RexTM (HEK293T)-TMEM223$^{-/-}$ | This paper | N/A | Cell line generated as described in Materials and methods |
| Transfected construct (*H. sapiens*) | pX330-TMEM223 gRNA | This paper | N/A | Cloning described in Materials and methods |
| Transfected construct (*H. sapiens*) | pEGFPN1 | CloneTech | N/A | |
| Transfected construct (*H. sapiens*) | pCDNA5-mL62-FLAG | This paper | N/A | Construct generated by mutagenesis of pCDNA5-mL62-FLAG |
| Transfected construct (*H. sapiens*) | pCDNA5-MITRAC7-FLAG | *Dennerlein et al., 2015* (Cell Rep.) | N/A | Construct generated by mutagenesis of pCDNA5-MITRAC7-FLAG |
| Transfected construct (*H. sapiens*) | pCDNA5-C12ORF62-FLAG | *Richter-Dennerlein et al., 2016* (Cell) | N/A | Construct generated by mutagenesis of pCDNA5-C12ORF62-FLAG |
| Transfected construct (*H. sapiens*) | pCDNA5-MITRAC12- FLAG | *Aich et al., 2018* (eLife) | N/A | Construct generated by mutagenesis of pCDNA5-MITRAC12-FLAG |
| Transfected construct (*H. sapiens*) | pCDNA5-C12ORF73- FLAG | This paper | N/A | Construct generated by mutagenesis of pCDNA5-C12ORF73-FLAG |
| Transfected construct (*H. sapiens*) | pCDNA5-SMIM4-FLAG | This paper | N/A | Construct generated by mutagenesis of pCDNA5-SMIM4-FLAG |
| Transfected construct (*H. sapiens*) | pCDNA5-UQCC1-FLAG | This paper | N/A | Construct generated by mutagenesis of pCDNA5-UQCC1-FLAG |
| Transfected construct (*H. sapiens*) | pCDNA5-UQCC2-FLAG | This paper | N/A | Construct generated by mutagenesis of pCDNA5-UQCC2-FLAG |
| Transfected construct (*H. sapiens*) | pCDNA5-UQCC3-FLAG | This paper | N/A | Construct generated by mutagenesis of pCDNA5-UQCC3-FLAG |
| Antibody | TMEM223 rabbit polyclonal | Self made | PRAB4850 | (1:500) |
| Antibody | SMIM4 rabbit polyclonal | Self made | PRAB5494 | (1:500) |
| Antibody | mS40 (MRPS18B) rabbit polyclonal | ProteinTech | RRID:AB_2146368 | (1:1000) |

*Continued on next page*

*Continued*

| Reagent type (species) or resource | Designation | Source or reference | Identifiers | Additional information |
|---|---|---|---|---|
| Antibody | TMEM177 rabbit polyclonal | Self made | PRAB4988 | (1:1000) |
| Antibody | TIM44 rabbit polyclonal | Self made | PRAB5142 | (1:4000) |
| Antibody | COX6A rabbit polyclonal | Self made | PRAB3282 | (1:1000) |
| Antibody | SLIRP rabbit polyclonal | Self made | PRAB3813 | (1:500) |
| Antibody | CYTB rabbit polyclonal | Self made | PRAB5151 | (1:1000) |
| Antibody | C12ORF73 rabbit polyclonal | Self made | PRAB5105 | (1:500) |
| Antibody | PHB1 rabbit polyclonal | ProteinTech | RRID:AB_2164476 | (1:1000) |
| Antibody | uL3m rabbit polyclonal | ProteinTech | RRID:AB_10639509 | (1:1000) |
| Antibody | uS14m rabbit polyclonal | ProteinTech | RRID:AB_2878240 | (1:2000) |
| Antibody | bS16m rabbit polyclonal | ProteinTech | RRID:AB_2180166 | (1:5000) |
| Antibody | YME1L rabbit polyclonal | Self made | PRAB5113 | (1:500) |
| Antibody | SLP2 rabbit polyclonal | ProteinTech | RRID:AB_2286822 | (1:1000) |
| Antibody | AFG3L2 rabbit polyclonal | Self made | PRAB5149 | (1:500) |
| Antibody | MITRAC12 rabbit polyclonal | Self made | PRAB3761 | (1:1000) |
| Antibody | C12ORF62 rabbit polyclonal | Self made | PRAB4844 | (1:500) |
| Antibody | MITRAC7 rabbit polyclonal | Self made | PRAB4843 | (1:500) |
| Antibody | COX1 rabbit polyclonal | Self made | PRAB2035 | (1:2000) |
| Antibody | COX4-1 rabbit polyclonal | Self made | PRAB1522 | (1:2000) |
| Antibody | uL23m rabbit polyclonal | Self made | PRAB1716 | (1:500) |
| Antibody | uL1m rabbit polyclonal | Self made | PRAB4969 | (1:500) |
| Antibody | TOM70 rabbit polyclonal | Self made | PRAB3280 | (1:1000) |
| Antibody | TACO1 rabbit polyclonal | Self made | PRAB3627 | (1:500) |
| Antibody | MITRAC15 rabbit polyclonal | Self made | PRAB4814 | (1:500) |
| Antibody | FLAG rabbit polyclonal | Sigma-Aldrich | RRID:AB_259529 | (1:2000) |
| Antibody | TIM21 rabbit polyclonal | Self made | PRAB3674 | (1:2000) |
| Antibody | SDHA Mouse monoclonal | Self made | PRAB4978 | (1:2000) |
| Antibody | Rieske rabbit polyclonal | Self made | PRAB1512 | (1:2000) |
| Antibody | ATP5B rabbit polyclonal | Self made | PRAB4826 | (1:10000) |
| Antibody | TIM44 rabbit polyclonal | ProteinTech | RRID:AB_2204679 | (1:2500) |
| Antibody | NDUFB8 rabbit polyclonal | Self made | PRAB3764 | (1:500) |

*Continued on next page*

*Continued*

| Reagent type (species) or resource | Designation | Source or reference | Identifiers | Additional information |
|---|---|---|---|---|
| Antibody | NDUFA9 rabbit polyclonal | Self made | PRAB1524 | (1:500) |
| Antibody | TIM23 rabbit polyclonal | Self made | PRAB1527 | (1:2000) |
| Antibody | SCO2 rabbit polyclonal | Self made | PRAB4982 | (1:500) |
| Antibody | FAM36A rabbit polyclonal | Self made | PRAB4490 | (1:500) |
| Antibody | SURF1 rabbit polyclonal | Self made | PRAB1528 | (1:1000) |
| Recombinant DNA reagent | QuikChange Site-Directed Mutagenesis Kit | Agilent | 210515 | |
| Recombinant DNA reagent | KOD Hot Start DNA Polymerase | Merck | 71086-3 | |
| Recombinant DNA reagent | First Strand cDNA Synthesis Kit | Thermo Fisher Scientific | K1612 | |
| Commercial assay or kit | Human Complex IV Activity Kit | Abcam | ab109910 | |
| Chemical compound, drug | GeneJuice | Merck | 70967-3 | |
| Chemical compound, drug | Anti-FLAG M2 Affinity Gel | Sigma-Aldrich | A2220 | |
| Chemical compound, drug | TRIzol | Thermo Fisher Scientific | 15596026 | |
| Chemical Compound, drug | Protein-A SepharoseTM CL-4B | GE Healthcare | 17-0963-03 | |
| Chemical Compound, drug | [35 S]methionine | Hartmann Analytic | SCM-01 | |
| Chemical compound, drug | Emetine dihydrochloride hydrate | Sigma-Aldrich | 219282 | |
| Software, algorithm | ImageQuantTL 7.0 software | GE Healthcare | RRID:SCR_014246 | |
| Software, algorithm | ImageJ 1.47v | NIH | RRID:SCR_003070 | |
| Software, algorithm | Geneious | Biomatters Ltd | RRID:SCR_010519 | |
| Software, algorithm | Prism5 | GraphPad Software | RRID:SCR_015807 | |

## Cell culture of HEK293T cells

Human embryonic kidney cell lines (HEK293-Flp-In T-Rex; HEK293T; Thermo Fisher Scientific) were cultured either in high glucose (4.5 mg/ml) or galactose (0.9 mg/ml) containing Dulbecco's modified Eagle's medium (DMEM) media; supplemented with 10% (v/v) fetal bovine serum (FBS) (Capricorn Scientific), 1 mM sodium pyruvate, 2 mM L-glutamine, and 50 µg/ml uridine at 37°C under a 5% $CO_2$ humidified atmosphere. Cell counts were performed by using the Neubauer counting chamber. For inhibition of cytosolic translation, DMEM was supplemented with 100 µg/ml emetine dihydrochloride hydrate (Sigma-Aldrich). Cells for SILAC analysis were cultured as previously described (*Mick et al., 2012*).

Cell lines were regularly tested for mycoplasma contaminations (Eurofins Genomics) and passage. HEK293T cell lines expressing SMIM4[FLAG] (NM_001124767) and C12ORF73[FLAG] (NM_001135570.3) under the control of a tetracycline-inducible CMV promotor or UQCC1[FLAG] (NM_001184977.2), UQCC2[FLAG] (NM_032340.4), and UQCC3[FLAG] (NM_001085372.3) (non-inducible) were generated as described previously (*Mick et al., 2012*).

The knockout cell line of TMEM223 was generated as described previously (*Aich et al., 2018*). Briefly, the TMEM223 (NM_001080501.3) specific oligonucleotides GCAAGGCACGACGCTGCAAC and its reverse complement were cloned into the pX330 vector and co-transfected with the pEGFP-N1 plasmid into HEK293T WT cells. Three days after transfection, single cells were sorted by flow cytometry. Single colonies were screened by immunoblotting and sequencing of the corresponding gene region.

To inhibit mitochondrial translation WT cells were treated with thiamphenicol (TAP) with a final concentration of 50 µg/ml for 48 hr.

## Cultivation and STED super-resolution light microscopy of HEK293T cells

HEK293T cells were cultivated in DMEM, containing 4.5 g/L Glucose and GlutaMAX additive (Thermo Fisher Scientific, Waltham, MA) supplemented with 1 mM sodium pyruvate (Sigma-Aldrich, St. Louis, MO/Merck Millipore, Burlington, MA) and 10% (v/v) FBS (Merck Millipore, Burlington, MA).

Prior microscopy, cells were cultivated on glass cover slides for 1–3 days at 37°C and 5% $CO_2$. Expression of SMIM4[FLAG] and C12ORF73[FLAG] was induced by incubation with 1 µg/ml Doxycycline-hyclate (Sigma-Aldrich, St. Louis, MO/Merck Millipore, Burlington, MA) for 24 hr. Fixation and labeling were done as described previously (*Wurm et al., 2010*). Essentially, cells were fixed using an 8% (w/v) formaldehyde solution, permeabilized by incubation with a 0.25% (v/v) Triton X-100 solution, and blocked with a 5% (w/v) bovine serum albumin (BSA) solution.

The investigated proteins were labeled by antisera specific for TOM22 (Anti-Tom22, Rabbit, Merck Millipore/Sigma-Aldrich, HPA003037, Lot: B119406) and the FLAG-tag (Anti-FLAG, Mouse, Merck Millipore/Sigma-Aldrich, clone M2, F3165, Lot: SLBQ7119V), respectively. Detection was accomplished via secondary antibodies custom labeled with the dyes ALEXA Fluor594 (Thermo Fisher Scientific, Waltham, MA) or Abberior STAR RED (Abberior, Göttingen, Germany), respectively. DNA was labeled via Quant-iT PicoGreen dsDNA reagent (Thermo Fisher Scientific, Waltham, MA). The samples were mounted using Mowiol containing 1,4-Diazabicyclo[2.2.2]octan (DABCO). STED images were acquired using a quad scanning STED microscope (Abberior Instruments, Göttingen, Germany) equipped with a UPlanSApo 100×/1.40 Oil objective (Olympus, Tokyo, Japan). For excitation of the respective dyes, laser beams featuring wavelengths of 485 nm, 561 nm, and 640 nm were used. STED was performed applying a laser beam with a wavelength of 775 nm. For all images, a pixel size of 15 nm was utilized. Light microscopy data were linearly deconvolved using the Imspector software (Abberior Instruments).

## siRNA constructs and application

To generate knockdown cells, HEK293T WT cells were transiently transfected with siRNA oligonucleotides against SMIM4 (5′-GCA-GUC-AAU-AAA-GUC-AAU-A-3′) and C12orf73 (5′-ACA-CAA-ACC-UCA-AGU-UUC-U-3′) to a final concentration of 33 nM and non-targeting siRNA used as control (Eurogentec). siRNA targeting TMEM223 (GUU-CCU-UUG-AAG-CAG-GUA-U) was used in a final concentration of 16 nM (non-targeting control accordingly). Lipofectamine RNAi-MAX (Invitrogen) was used as transfection reagent, following the manufacturer's protocol. Afterward, cells were incubated at 37°C under 5% $CO_2$ atmosphere for 72 hr.

## [35S]methionine labeling of newly synthesized mitochondrial-encoded proteins

HEK293T cells were starved with FCS/methionine-free media. After inhibition of cytosolic translation by adding 100 µg/ml emetine dihydrochloride hydrate (Sigma-Aldrich), cells were incubated for 1 hr with [35S]methionine in a concentration of 0.2 mCi/ml in fully supplemented DMEM without methionine. In case of puromycin treatment, samples were pulsed-labeled with [35S]methionine for 10 min before addition of 2 µg/ml puromycin for another 20 min incubation step.

Subsequently, cells were harvested and washed with PBS, and used for further analysis on SDS-PAGE and autoradiography. Radioactive signals could be detected on Storage Phosphor Screens via a Typhoon FLA 7000 scanner (GE Healthcare) after several days of incubation.

## Cytochrome *c* oxidase activity assay

The activity of the cytochrome *c* oxidase was measured as described previously (*Dennerlein et al., 2015*). A cytochrome c oxidase specific activity microplate assay kit (Mitosciences, Abcam) was used, following the manufacturer's protocol. The specific activity of the cytochrome *c* oxidase was measured according to the manufacturer's instructions. In general, 15 mg of cell lysate was used per well. The oxidization of cytochrome *c* was measured at 550 nm, representing cytochrome *c* oxidase activity.

## Isolation of mitochondria

Mitochondria were isolated according to a modified protocol (*Panov and Orynbayeva, 2013*). Cells were harvested using PBS and resuspended in cold TH-buffer (300 mM Trehalose, 10 mM KCl, 10 mM

HEPES; pH7.4) with 2 mM PMSF and 0.1 mg BSA/ml. Subsequently, cells were gently homogenized two times using a Potter S Dounce Homogenizer (Sartorius) and pelleted at 400×*g* for 10 min at 4°C after each homogenization step. The supernatant was collected and remaining cell debris were removed by additional centrifugation (800×*g*, 8 min, 4°C). Afterward, mitochondria were pelleted at 11,000×*g* for 10 min at 4°C and pooled mitochondria pellets were washed with BSA-free TH-buffer and collected by centrifugation as described before. Finally, mitochondria were resuspended in BSA-free TH buffer and stored at –80°C or used right away.

## Protein localization and protease protection assays

Carbonate extraction and mitochondrial swelling experiments were performed as previously described (*Mick et al., 2012*). For carbonate extraction, mitochondria were isolated as described and resuspended in buffer containing 10 mM 3-(N-morpholino) propanesulfonic acid (MOPS) (pH 7.2), 50 mM NaCl and either 1% Triton X-100 or 0.1 M carbonate at pH 10.5 or 11.8. Insoluble membranes were pelleted by 55,000 rpm at 4°C, 45 min in a TLA-55 rotor (Beckman Coulter). For submitochondrial localization, mitochondria were either suspended in SEM buffer (250 mM sucrose, 1 mM EDTA, and 10 mM MOPS [pH 7.2]), to osmotically stabilize mitochondria, or in EM buffer (1 mM EDTA, and 10 mM MOPS [pH 7.2]), to rupture the outer mitochondrial membrane. In the following, Proteinase K (PK) was added as indicated. Furthermore, mitochondria were lysed with 1% Triton X-100 in the presence of PK for positive control. All reactions were stopped after 10 min by addition of PMSF (2 mM final concentration), followed by trichloroacetic acid (TCA) precipitation.

## Affinity purification of protein complexes

Isolated mitochondria or cells were lysed in solubilization buffer (150 mM NaCl, 10% glycerol (v/v), 20 mM MgCl2, 2 mM PMSF, 50 mM Tris-HCl, pH 7.4, 1% digitonin (v/w) protease inhibitor) in a ratio of 1–2 µg/µl for 30 min at 4°C and 850 rpm. Lysates were cleared by centrifugation (15 min, 16,000×*g*, 4°C) and transferred onto anti-FLAG M2 agarose beads (Sigma-Aldrich) for FLAG immunoprecipitation. After 1 hr binding at 4°C, beads were washed several times (10×) with washing buffer (50 mM Tris-HCl, pH 7.4, 150 mM NaCl, 10% glycerol (v/v), 20 mM MgCl$_2$, 1 mM PMSF, 0.3% digitonin (v/w)) to remove unbound proteins. Bound proteins were eluted with FLAG peptide (Sigma-Aldrich) by a 30 min incubation step at 850 rpm at 4°C. Samples were analyzed via SDS-PAGE and immunoblotting or quantitative mass spectrometry.

For antibody immunoprecipitation, same protocol was used as described above. Lysed mitochondria or cells were transferred onto protein A-Sepharose (PAS) containing crosslinked SMIM4 or C12orf73 antibody in a Mobicol spin column (MoBiTec). Bound proteins at PAS-anti SMIM4/C12orf73 columns were eluted by adding 0.1 M glycine, pH 2.8 for 6 min at 650 rpm at room temperature (RT).

## Immunoblotting via western blot

Proteins were separated using an SDS-PAGE and afterward transferred onto PVDF membrane (Millipore) by semidry blotting (Blotting buffer). Primary antibodies were incubated overnight at 4°C or 1 hr at RT. Secondary antibodies (rabbit or mouse) were incubated at RT for additional 1–2 hr. Signals were visualized on X-ray films using the enhanced chemiluminescence detection kit (GE Healthcare), and quantifications were performed using ImageQuant TL 7.0 software (GE Healthcare) with a rolling ball background subtraction.

## Blue native and second dimension analysis

BN-PAGE was performed as described previously (*Mick et al., 2012*). Isolated mitochondria or cells were solubilized in a concentration of 1 µg/µl in BN-PAGE lysis buffer containing 1% digitonin or 1% DDM (20 mM Tris-HCl, pH 7.4, 0.1 mM EDTA, 50 mM NaCl, 10% glycerol (v/v), 1 mM PMSF). After a 20-min incubation step on ice, debris was removed by centrifugation for 15 min, 16.000×*g* at 4°C. The remaining supernatant was resuspended in BN-PAGE Loading dye (5% Coomassie Brilliant Blue G250 (v/w), 500 mM 6-aminocaproic acid, 100 mM Bis-Tris–HCl, pH 7.0) and applied to electrophoresis on a 4–13% or 2.5–10% gradient gel. Afterward, proteins were either transferred to a PVDF membrane by western blot method or subjected to 2D-PAGE analysis. In case of 2D-PAGE analysis, signal BN-PAGE stripes were cut out from gel and further used in SDS-PAGE separation.

## Quantitative mass spectrometry and data analysis

Affinity-purified SILAC-labeled mL62[FLAG] and SMIM4[FLAG] complexes were processed for quantitative LC-MS analysis following a gel-based approach. Reduction and alkylation of cysteine residues and subsequent tryptic in-gel digestion of proteins were performed as described before (*Peikert et al., 2017*). Peptides were desalted using StageTips, dried in vacuo, and reconstituted in 0.1% trifluoroacetic acid. LC-MS analyses were carried out using an UltiMate 3000 RSLCnano HPLC system (Thermo Fisher Scientific) coupled to an Orbitrap Elite mass spectrometer (Thermo Fisher Scientific). Peptides were separated on a C18 reversed-phase nano LC column (for mL62[FLAG] samples: Acclaim PepMap, 500 mm × 75 µm, 2 µm particle size, 100 Å packing density [Thermo Fisher Scientific], flow rate of 0.25 µl/min; for SMIM4[FLAG] samples: nanoEase M/Z HSS C18 T3, 250 mm × 75 µm, 1.8 µm particle size, 100 Å packing density [Waters], flow rate of 0.3 µl/min) using a binary solvent system consisting of 4% dimethyl sulfoxide/0.1% formic acid (solvent A) and 48% methanol/30% acetonitrile/4% dimethyl sulfoxide/0.1% formic acid (solvent B). The gradients employed for peptide elution were as follows: 1% solvent B for 5 min, 1–65% B in 50 min, 65–95% B in 5 min, 5 min at 95% B for mL62[FLAG] samples and 7% solvent B for 5 min, 7–65% B in 65 min, 65–80% B in 5 min, 5 min at 80% B for SMIM4[FLAG] samples.

The Orbitrap Elite was operated in a data-dependent mode. MS survey scans were acquired at a mass range of *m/z* 370–1700 and a resolution of 120,000 (at *m/z* 400). The target value was $10^6$ ions and the maximum injection time was 200 ms. Up to 15 (mL62[FLAG] complexes) or 25 (SMIM4[FLAG] complexes) of the most intense multiply charged peptide ions were selected for fragmentation by collision-induced dissociation in the linear ion trap at a normalized collision energy of 35%, an activation q of 0.25, and an activation time of 10 ms. The target value was set to 5000 ions, the maximum injection time to 150 ms, the isolation width to 2.0 *m/z*, and the dynamic exclusion time to 45 s.

MS raw data were analyzed with MaxQuant/Andromeda (version 1.4.1.2 for mL62[FLAG] and 1.5.5.1 for SMIM4[FLAG] data; *Cox and Mann, 2008*; *Cox et al., 2011*) and searched against the UniProt human proteome set including isoforms (release versions 08/2014 for mL62[FLAG] and 08/2018 for SMIM4[FLAG] data) using default settings except that the minimum requirements for protein identification and relative quantification were set to one unique peptide and one SILAC peptide pair, respectively. Arg10 and Lys8 were set as heavy labels. Carbamidomethylation of cysteine residues was considered as fixed, and N-terminal acetylation and oxidation of methionine as variable modifications. The options 'match between runs' and 'requantify' were enabled.

Experiments were performed in four biological replicates including label-switch. Lists of proteins identified in the analyses of mL62[FLAG] and SMIM4[FLAG] complexes are provided in *Supplementary file 1* and *Supplementary file 2*, respectively.

## Quantifications and statistical analysis

To measure signals for quantifications, the ImageQuant software of GE Healthcare was used. For statistical analysis, a one-sample t-test (https://www.graphpad.com) was used. Statistics were defined as *≤0.05, **≤ 0.01, and *** ≤ 0.001.

## Acknowledgements

The authors thank B Knapp, C Ronsör, and F Holtkotte for technical assistance. The authors like to thank Nils-Göran Larsson for open discussions of unpublished data. Supported by the European Research Council Advanced grants (ERCAdG No. 835102, SJ) and (ERCAdG No. 339580, PR), Deutsche Forschungsgemeinschaft (DFG) under Germany's Excellence Strategy – EXC 2067/1-390729940 to PR, SJ, RRD, the DFG-funded FOR2848 (project P01, TL; P04, SJ & PR), SFB1190 (P13, PR), the DFG-funded Emmy-Noether grant (RI 2715/1-1, RRD). BW is supported by the European Research Council Consolidator Grant 648235 and the Deutsche Forschungsgemeinschaft (DFG) Project 403222702/SFB 1381 and Germany's Excellence Strategy (CIBSS–EXC-2189 – Project ID 390939984).

## Additional information

### Funding

| Funder | Grant reference number | Author |
|---|---|---|
| European Research Council | ERCAdG No. 835102 | Stefan Jakobs |
| European Research Council | ERCAdG No. 339580 | Peter Rehling |
| Deutsche Forschungsgemeinschaft | EXC 2067/1-390729940 | Peter Rehling Ricarda Richter-Dennerlein |
| Deutsche Forschungsgemeinschaft | EXC2067/1-390729940 | Stefan Jakobs |
| Deutsche Forschungsgemeinschaft | FOR2848 | Peter Rehling Thomas Langer Stefan Jakobs |
| Deutsche Forschungsgemeinschaft | SFB1190 | Peter Rehling |
| Deutsche Forschungsgemeinschaft | RI 2715/1-1 | Ricarda Richter-Dennerlein |
| European Research Council | Consolidator Grants 648235 | Bettina Warscheid |
| Deutsche Forschungsgemeinschaft | 403222702/SFB1381 | Bettina Warscheid |
| Deutsche Forschungsgemeinschaft | EXC 2189 390939984 | Bettina Warscheid |
| Max Planck Institute for Dynamics of Complex Technical Systems Magdeburg | | Peter Rehling |

The funders had no role in study design, data collection and interpretation, or the decision to submit the work for publication.

### Author contributions

Sven Dennerlein, Conceptualization, Data curation, Methodology, Supervision, Validation, Writing – original draft, Writing – review and editing; Sabine Poerschke, Conceptualization, Data curation, Investigation, Methodology, Writing – original draft, Writing – review and editing; Silke Oeljeklaus, Data curation, Methodology, Software, Writing – review and editing; Cong Wang, Johannes Sattmann, Diana Bauermeister, Elisa Hanitsch, Data curation, Methodology; Ricarda Richter-Dennerlein, Data curation, Methodology, Supervision, Writing – review and editing; Stefan Stoldt, Data curation, Methodology, Writing – review and editing; Thomas Langer, Writing – review and editing; Stefan Jakobs, Methodology, Visualization, Writing – review and editing; Bettina Warscheid, Data curation, Writing – review and editing; Peter Rehling, Conceptualization, Funding acquisition, Supervision, Writing – original draft, Writing – review and editing

### Author ORCIDs

Sven Dennerlein http://orcid.org/0000-0002-8901-0242
Stefan Jakobs http://orcid.org/0000-0002-8028-3121
Bettina Warscheid http://orcid.org/0000-0001-5096-1975
Peter Rehling http://orcid.org/0000-0001-5661-5272

### Decision letter and Author response

Decision letter https://doi.org/10.7554/eLife.68213.sa1
Author response https://doi.org/10.7554/eLife.68213.sa2

## Additional files

### Supplementary files
• Supplementary file 1. Mass spectrometric analyses of mL62 and SMIM4 containing complexes. Isolated complexes of mL62FLAG (*Supplementary file 1*) or from SMIM4FLAG (*Supplementary file 2*) were subjected to quantitative mass spectrometric analyses (compare *Figures 1B and 4E*). MS raw data were analyzed with MaxQuant/Andromeda (see Materials and methods section). Analyzed datasets are presented as Excel files.

• Supplementary file 2. Mass spectrometric analyses of mL62 and SMIM4 containing complexes.

• Transparent reporting form

### Data availability
All data generated during this study are included in the manuscript figures.

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
