## [Editor Report]

In this work, the authors analyze the interactome of the human mitochondrial ribosomes and identify two new mitochondrial inner membrane proteins, TMEM223 and SMIM4, as ribosome-associated proteins involved in the biogenesis of respiratory chain complexes. The study reveals novel assembly factors for complex III and IV biogenesis that link early assembly stages to the mitochondrial translation machinery.

---

## [Decision Letter]

**Decision letter after peer review:**

Thank you for submitting your article "Defining the interactome of the human mitochondrial ribosome identifies SMIM4 and TMEM223 as respiratory chain assembly factors" for consideration by *eLife*. Your article has been reviewed by 3 peer reviewers, and the evaluation has been overseen by a Reviewing Editor and David Ron as the Senior Editor. The following individual involved in review of your submission has agreed to reveal their identity: Andre Schneider (Reviewer #3).

The reviewers are convinced that the work is interesting, important and of high technical quality. They have discussed their reviews with one another, and the Reviewing Editor has drafted this to help you prepare a revised submission.

Essential revisions:

1. TMEM 223 and its relation to Cox1 and its partners should be further clarified, including Cmc1. At which stage is TMEM223 released from the intermediates? An important control with TMEM223 complementation in the KO cells is needed in critical experiments to exclude offtargets.

2. What underlies an enhancement of the CIII dimer in the absence of TMEM223 – an increase in the subunits, stability or binding to CIV?

3. Concerning C12ORF73 and SMIM4, a more detailed analysis should be added to understand better their role, specifically in the light of discrepancies with Zhang et al., 2020. This analysis should include addressing the interaction with Core1.

Furthermore in the specific comments, the referees highlighted several points that must be addressed. Especially better presentation of the literature and its discussion as well as introducing quantifications, whenever possible, and statistical significance, will improve the clarity of the findings. Please referrer to the section appended below to see the list of detailed corrections- some of them may repeat.

– The manuscript omitted a relevant paper that must be discussed in the context of this work, concerning the updated CIII assembly pathway that includes the precise dimerization step in yeast (doi: 10.1016/j.bbabio.2020.148177). There is also need to update several citations referring to CIV assembly (doi: 10.1016/j.semcdb.2017.08.055; doi: 10.7554/*eLife*.32572), as well as to mitochondrial translation (doi: 10.1016/j.tcb.2017.05.004) and to mitoribosome assembly (doi: 10.1016/j.tcb.2020.12.008; doi: 10.1002/1873-3468.14024).

– To allow reproducibility, the authors must detail in the Methods section the commercial sources and catalogue numbers of the antibodies used in this work.

– The authors must include a table describing the reference Genbank sequences used for the guide RNA sequence selection in TMEM223-/- cells, as well as the precise genomic mutations (indels) generated by CRISPR-Cas9-mediated genome editing in the different alleles and their expected impacts on mRNA and protein sequences.

– As a general comment, the mitoribosome interactome includes many assembly factors in addition to the two studied, such as the known COX20 (early COX2 assembly line) or COA1 (not so early COX1 assembly line). The authors could comment on whether this indicates that respiratory complex assembly starts on the mitoribosome platform and continues until a particular assembly stage for each complex.

– MITRAC12-C12ORF62 are frequently referred to in the literature as COA3 and COX14. The alias names should be stated to allow the reader to follow the manuscript better.

– TMEM223 is not essential for COX assembly. Does it play an overlapping role with any other assembly factor?

– Regarding SMIM4 and C12ORF73, the silencing experiments seem to indicate that the factors are not essential for CIII assembly. This concept could be further clarified.

– Major suggestion is that the authors should present quantifications of some experiments.

It is claimed for Figure 2D that COX4I1, COX6A, C12ORF62 and MITRAC12 are unaffected in TMEM223 minus cells. This is very difficult to see especially for C12ORF62 and MITRAC12. The levels of NDUFA9, SDHA, RIESKE, and ATP5B on the other hand are classified as constant. Again sometimes it is difficult to see that they behave differently to the first set of proteins. Densitometric scanning of triplicate blots could help here.

Same goes for Figure 3A it would make sense to quantify the extent of assembly inhibition in the TMEM223 minus cells, as has been done for Figure 3C. It would also be nice to see a loading control for the BN-PAGE gels, the Coomassie stain would be sufficient for this.

It seems to me there is an increase in the assembly of CYTB, this should be discussed.

Also the BN-PAGE blots in Figure 5D and 5E should be quantified. Especially for 5E it would help to know which complex is most affected. Moreover, it should be discussed why ablation of C12ORF73, in contrast to SMIM4, besides complex III also affects complex I and IV.*Reviewer #1:*

In this work, Dennerlein et al., dissect the interactome of the human mitochondrial ribosomes in HEK293T cells and identify two new mitochondrial inner membrane proteins, TMEM223 and SMIM4, as ribosome-associated proteins involved in the biogenesis of respiratory chain complexes IV and III, respectively. While TMEM223 seems to preferentially stimulate translation of COX1 mRNA through its interaction with early COX1 assembly intermediates, SMIM4 interacts with newly synthesized cytochrome b to support initial steps of complex III biogenesis in concurrence with three previously identified CIII assembly factors: UQCC1, UQCC2 and C12ORF73. These analyses thus reveal novel assembly factors for complex III and IV biogenesis that link early assembly stages to the mitochondrial translation machinery, and expand the number of genes potentially involved in mitochondrial diseases. Overall, the data presented in this manuscript are of high quality and the manuscript is well written. The new observations are extremely interesting, and add relevant new value compared with previous reports.

*Reviewer #2:*

In this study, Dennerlein and colleagues investigate the broad mitochondrial ribosome interactome. Among the many proteins identified, the authors focus on two new ribosome-associated proteins, SMIM4 and TMEM223, which are found to function as early assembly factors for the mitochondrial respiratory chain complexes III and IV, respectively. Specifically, the manuscript collects in a well-organized and logical sequence substantial experimental evidence supporting the role of TMEM223 in early COX1 assembly and of SMIM4, together with C12ORF73, in the assembly of CYTB. These data add to the fields of mitochondrial respiratory complex biogenesis and the coordination of mitochondrial translation with respiratory complex assembly.

*Reviewer #3:*

The manuscript of Dennerlein et al., identifies that two human integral mitochondrial inner membrane proteins, TMEM233 and SMIM4, which are associated with the large mitoribosomal subunit, as early assembly factors for cytochrome c oxidase (complex IV) or and cytochrome c reductase (complex III), respectively. TMEM233 is required for efficient translation of mitochondrially encoded Cox1 and associated with previously characterized early assembly intermediates of cytochrome c oxidase. SMIM4 is associated with C12ORF73, which has previously been implicated in cytochrome c reductase assembly, and downregulation of either of the two proteins results in destabilization of the other. Both ablation of either SIMM4 and C12ORF73 primarily affects assembly of complex III. This is in line with the observation that both SIMM4 and C12ORF73 interact with previously characterized early assembly factors of complex III but not with later ones. Finally, both proteins associate with the newly synthesize cytochrome b.

The manuscript is well written, has clear figures and the results are presented in a logical way. I find the approach of identifying early assembly factors by characterizing the interactome of the large mitoribosomal subunit an elegant one. The presented results are of high quality, convincing and support the conclusions drawn be the authors. Essentially all claims made by the authors are supported by more than one independent experimental approach. The only issue I have is that the phenotypes of the TMEM233 and SMIM4 depleted cells could be described in more quantitative terms.

---

## [Author Response]

Essential revisions:1. TMEM 223 and its relation to Cox1 and its partners should be further clarified, including Cmc1. At which stage is TMEM223 released from the intermediates? An important control with TMEM223 complementation in the KO cells is needed in critical experiments to exclude offtargets.

As suggested, we investigated the interaction of TMEM223 (including CMC1) in more detail. The results are presented as new Figures 3 D, E, F and G.

Our analyses show that physiological levels of TMEM223 are crucial and that the expression systems that were available to us were insufficient to fine tune appropriate steady state levels of TMEN223 in the knockout cell line. Therefore, to support our findings in the knockout cell lines, we applied siRNA-mediated depletion of TMEM223. As presented in the new Figure 3C, siRNA mediated reduction of TMEM223 led to a reduction of COX1 translation comparable to the TMEM223^-/-^ cell line. These findings support our previous observations.

2. What underlies an enhancement of the CIII dimer in the absence of TMEM223 – an increase in the subunits, stability or binding to CIV?

As suggested, we investigated the steady state levels of the complex III RIESKE subunit in TMEM223-/- cells (Figure 2D with quantification) and complex III assembly factors (new Figure 2 —figure supplement 1 C). We assessed the stability of RIESKE, UQCC1 and C12ORF73 (new Figure 2 —figure supplement 1 D).

3. Concerning C12ORF73 and SMIM4, a more detailed analysis should be added to understand better their role, specifically in the light of discrepancies with Zhang et al., 2020. This analysis should include addressing the interaction with Core1.

To address this, we performed immunoisolations of UQCRQ and RIESKE. The data has been incorporated as new Figures 5 —figure supplement 2 D and E supporting our conclusions.

Furthermore in the specific comments, the referees highlighted several points that must be addressed. Especially better presentation of the literature and its discussion as well as introducing quantifications, whenever possible, and statistical significance, will improve the clarity of the findings. Please referrer to the section appended below to see the list of detailed corrections- some of them may repeat.– The manuscript omitted a relevant paper that must be discussed in the context of this work, concerning the updated CIII assembly pathway that includes the precise dimerization step in yeast (doi: 10.1016/j.bbabio.2020.148177). There is also need to update several citations referring to CIV assembly (doi: 10.1016/j.semcdb.2017.08.055; doi: 10.7554/eLife.32572), as well as to mitochondrial translation (doi: 10.1016/j.tcb.2017.05.004) and to mitoribosome assembly (doi: 10.1016/j.tcb.2020.12.008; doi: 10.1002/1873-3468.14024).

We have added references as suggested.

– To allow reproducibility, the authors must detail in the Methods section the commercial sources and catalogue numbers of the antibodies used in this work.

As suggested, we provide a resource table, according *eLife* standards.

– The authors must include a table describing the reference Genbank sequences used for the guide RNA sequence selection in TMEM223-/- cells, as well as the precise genomic mutations (indels) generated by CRISPR-Cas9-mediated genome editing in the different alleles and their expected impacts on mRNA and protein sequences.

As suggested, we added the Genbank number of TMEM223 and the guide sequences used to generate the knockout to the method section. Furthermore, we describe the generated mutations.

– As a general comment, the mitoribosome interactome includes many assembly factors in addition to the two studied, such as the known COX20 (early COX2 assembly line) or COA1 (not so early COX1 assembly line). The authors could comment on whether this indicates that respiratory complex assembly starts on the mitoribosome platform and continues until a particular assembly stage for each complex.

As suggested, we have extended the discussion to include this aspect.

– MITRAC12-C12ORF62 are frequently referred to in the literature as COA3 and COX14. The alias names should be stated to allow the reader to follow the manuscript better.

We agree with the Reviewer and added COA3 or COX14.

– TMEM223 is not essential for COX assembly. Does it play an overlapping role with any other assembly factor?

The reviewer raises an interesting point that we have addressed in the discussion.

– Regarding SMIM4 and C12ORF73, the silencing experiments seem to indicate that the factors are not essential for CIII assembly. This concept could be further clarified.

As suggested, we extended the discussion on that point, especially with regard to the potential link of C12ORF73 to cellular energy metabolism (Zhang et al., 2020).

– Major suggestion is that the authors should present quantifications of some experiments.It is claimed for Figure 2D that COX4I1, COX6A, C12ORF62 and MITRAC12 are unaffected in TMEM223 minus cells. This is very difficult to see especially for C12ORF62 and MITRAC12. The levels of NDUFA9, SDHA, RIESKE, and ATP5B on the other hand are classified as constant. Again sometimes it is difficult to see that they behave differently to the first set of proteins. Densitometric scanning of triplicate blots could help here.

As requested, we provide quantifications to improve the manuscript. We quantified steady state analysis of the TMEM223^-/-^ cell line and the results are now presented in a diagram, added to Figure 2D. We changed the text accordingly.

Same goes for Figure 3A it would make sense to quantify the extent of assembly inhibition in the TMEM223 minus cells, as has been done for Figure 3C. It would also be nice to see a loading control for the BN-PAGE gels, the Coomassie stain would be sufficient for thisIt seems to me there is an increase in the assembly of CYTB, this should be discussed.Also the BN-PAGE blots in Figure 5D and 5E should be quantified. Especially for 5E it would help to know which complex is most affected. Moreover, it should be discussed why ablation of C12ORF73, in contrast to SMIM4, besides complex III also affects complex I and IV.

As requested, we quantified BN-PAGE analyses. The quantification of former Figure 3A is now included as new Figure 2E and the corresponding diagrams of the quantifications of Figure 5D and 5E were added to those panels. Loading controls to each panel were added and are now presented as Figure 2 —figure supplement 1 A for Figure 2E (former Figure 3A); for Figure 5D and 5E as Figure 5 – supplement figure 2 F and G, respectively.

Furthermore, as mentioned above, we investigated the increase of assembly CYTB in more detail (new Figure 2 supplement figure 1 C and D) and discussed this.